# Dynamical mean-field theory for a highly heterogeneous neural population with graded persistent activity of the entorhinal cortex

**Futa Tomita** **, Jun-nosuke Teramae**\*

Graduate School of Informatics, Kyoto University, Kyoto, Japan

\* teramae@acs.i.kyoto-u.ac.jp

## Abstract

The entorhinal cortex serves as a major gateway connecting the hippocampus and neocortex, playing a pivotal role in episodic memory formation. Neurons in the entorhinal cortex exhibit two notable features associated with temporal information processing: a population-level ability to encode long temporal signals and a single-cell characteristic known as graded-persistent activity, where some neurons maintain activity for extended periods even without external inputs. However, the relationship between these single-cell characteristics and population dynamics has remained unclear, largely due to the absence of a framework to describe the dynamics of neural populations with highly heterogeneous time scales. To address this gap, we extend the dynamical mean field theory, a powerful framework for analyzing large-scale population dynamics, to study the dynamics of heterogeneous neural populations. By proposing an analytically tractable model of graded-persistent activity, we demonstrate that the introduction of graded-persistent neurons shifts the chaos-order phase transition point and expands the network's dynamical region, a preferable region for temporal information computation. Furthermore, we validate our framework by applying it to a system with heterogeneous adaptation, demonstrating that such heterogeneity can reduce the dynamical regime, contrary to previous simplified approximations. These findings establish a theoretical foundation for understanding the functional advantages of diversity in biological systems and offer insights applicable to a wide range of heterogeneous networks beyond neural populations.

## Author summary

Neurons in the brain exhibit a high degree of diversity in their intrinsic properties, including their characteristic time scales. However, little is known about how this diversity influences population dynamics. This study explores how a specific type of neuron

**Data availability statement:** The code used to run numerical simulations and analysis is

available at https://github.com/fuuta/multiD_hetero_DMFT.

**Funding:** F. T. was supported by Japan Science and Technology Agency (JST) (https://www.jst.go.jp/EN/), the establishment of university fellowships towards the creation of science technology innovation, Grant Number JPMJFS2123. J. T. is supported by Japan Society for the Promotion of Science (JSPS) (https://www.jsps.go.jp/english/) KAKENHI Grant Number 24K15104, 22K12186, and 23K21352, and Japan Agency for Medical Research and Development (AMED, https://www.amed.go.jp/en/index.html) AMED-CREST 23gm1510005h0003. The funders have no roles in the study design, data collection and analysis, decision to publish, or preparation of the manuscript.

**Competing interests:** The authors have declared that no competing interests exist.

in the entorhinal cortex, which can maintain firing activity for several minutes, even without external input, affects population dynamics. We develop a theory to describe large-scale recurrent networks of heterogeneous neurons and reveal that the introduction of these neurons shifts the network toward a more dynamic regime, which is preferable for temporal information processing. Our theory was also applied to other heterogeneous populations, offering new perspectives on the significance of diversity in neural population dynamics.

## Introduction

The entorhinal cortex, as the major gateway for information entering the hippocampus from various brain regions, plays an essential role in coding long temporal information, a crucial component of episodic memories [1–3]. Episodic memory is the ability of animals to store specific events they have experienced, along with their order and contextual details. The formation of such memories requires the brain to represent temporal information across various time scales. Recent experimental findings indicate that neurons in the lateral entorhinal cortex, along with hippocampal time cells [4–8], offer a fundamental mechanism for representing long temporal information required for episodic memory formation [9]. Experiments in rodents and humans have reported that neurons in the lateral entorhinal cortex exhibit activity characterized by gradually rising or decaying activity across various time scales [7,9]. A recent experiment also identified 'temporal context cells' in the entorhinal cortex, which, in contrast to time cells, respond rapidly following stimulus onset and then gradually return to baseline activity over extended temporal timescales, exhibiting a broad range of decay rates [10]. These activities may provide a suitable temporal code for the formation of episodic memory, capturing the different scales of time over which an animal's experiences occur [9,10].

In addition to the above population activity, the entorhinal cortex is known for a distinctive single-neuron activity, which is also associated with representing long-term information [11]. This activity is termed graded-persistent activity (GPA). The firing rate of an isolated neuron generally decays rapidly in the absence of external input. However, unlike the general response, some isolated entorhinal cortex neurons can sustain their firing activity for several minutes even after external inputs have ended [11]. Moreover, the sustained firing rate can exhibit various graded values, reflecting the input history of the neuron. Because this single-cell property suggests that a subset of neurons in the entorhinal cortex has much longer characteristic time scales compared to other typical neurons, it has been suggested that these neurons may be involved in functions requiring long temporal information, such as working memory [12–18].

Considering the above, one might expect that the existence of neurons with GPA could influence the population dynamics of the entorhinal cortex in encoding long-term temporal information. However, it remains unclear how the partial introduction of the GPA neurons modulates the dynamical properties of the neuronal population. The main cause of this difficulty is the lack of a theory to describe the dynamics of a highly heterogeneous population of neurons [19,20]. The fact that only a subset of entorhinal neurons exhibits extremely long-term graded-persistent activity implies that the intrinsic time scales of each neuron are largely diverse across the population, beyond the range where the heterogeneity can be considered negligible.

To address the problem and reveal how the partial introduction of GPA neurons modulates the population dynamics of neurons, we first propose an analytically tractable model of

neurons showing this characteristic activity. Then, we analyze their population dynamics by extending the dynamical mean-field theory (DMFT). DMFT is a comprehensive and efficient framework for analyzing the population dynamics of both biological and artificial recurrent neural networks [19,21–33]. The theory allows us to reduce the generally high-dimensional dynamics of large populations of neurons to an effective low-dimensional equation. This protocol is rigorously justified by a path integral approach. Specifically, the theory provides a theoretical basis for neural computation of temporal information, including temporal information processing in echo state networks and reservoir computing [34–36], by clarifying the onset of chaotic states in these networks.

DMFT enables us to marginalize the heterogeneous connection strengths between neurons into the effective mean field. However, we will see that we cannot simply average out the intrinsic properties of each neuron, including its characteristic time scale, by similarly incorporating them into the mean field. This is because of the difference in their dependency on the network size. To solve the problem, we extend the DMFT framework to networks consisting of heterogeneous neurons. Unlike conventional DMFT, which provides a single mean-field equation, we will obtain a set of mean-field equations reflecting the intrinsic heterogeneity of each neuron. Nevertheless, we will see that this set of equations can provide a single analytical expression to determine the critical coupling strength of the network. Results of the analysis show that the partial introduction of GPA neurons shifts the transition point to extend the dynamical region of the network. We confirm the validity of the theoretical prediction by comparing this with the results of numerical simulations with various network conditions.

To demonstrate the applicability of the approach, we will test the theory using networks with another type of heterogeneity, specifically the heterogeneity of adaptation in each neuron in the network, as discussed in a previous study [27]. The analysis in the previous work, based on the conventional treatment of heterogeneity, predicts that this heterogeneity would move the transition point in a way that expands the dynamical regime of the network. Contrary to this prediction, we will see that this heterogeneity can shift the network to stabilize the steady state, thus shrinking the dynamical region. This tendency is precisely described by the novel approach proposed here.

The organization of the paper is as follows. In the first subsection of the Results, an analytically tractable model of the GPA neuron is provided. Unlike previous models that have revealed underlying possible biological mechanisms of this characteristic activity [14,37–39], the model consists of only two variables. This model can describe the activity of a neuron with and without GPA by modulating a model parameter. Typical dynamics of networks of these neurons are also given in this subsection. DMFT of the heterogeneous network is developed in the next subsection. By deriving the equation determining the transition point, we discuss how adding GPA neurons to the network modulates its dynamics. Finally, we apply the proposed method to a network of neurons with heterogeneous adaptation. Possible extensions and remaining future subjects are examined in the Discussion. Details of the theory are given in the Methods section.

## Result

### A two-dimensional model for a neuron with and without graded-persistent activity

**Single-neuron model.**   A subset of neurons in the entorhinal cortex shows graded-persistent activity (GAP) characterized by prolonged sustained spike firings even after the input current to these neurons is terminated [11]. The sustained firing rate can be

continuously increased by repeated additions of depolarizing input current to the neuron and, conversely, can be gradually decreased by inductions of hyperpolarizing input. This unique feature is attributed to the single-cell property rather than network effects, as it remains even with the blockage of synaptic connections [11]. Previous studies have suggested the contribution of intracellular calcium density and membrane channels, such as the activation of calcium-dependent nonspecific cation channels, to the single-cell property, and several detailed computational models for this have been proposed [14,37–39].

We use a simple, analytically tractable two-dimensional model to study networks consisting of neurons with and without graded-persistent activity. The model consists of the variable $x$ representing the neural activity and an auxiliary variable $a$ whose time scale can be very slow. The firing rate of the neuron is given by an activation function $\phi(x)$ that is an increasing function of $x$. The auxiliary variable $a$ may correspond to some slow dynamics of the cell, such as intracellular calcium concentration. Depending on the value of a model parameter, the model can qualitatively describe both neurons with and without the GPA.

The single-cell model is given by

$$\dot{x}(t) = -x(t) + a(t) + I(t)$$
$$\dot{a}(t) = -\gamma a(t) + \beta x(t),$$

(1)

where the dot indicates the temporal derivative and $I(t)$ is the external input to the neuron. Coefficient $\gamma$ represents the decay rate of the auxiliary variable $a$, and $\beta$ is the feedback strength from the auxiliary variable $a$ to neural activity $x$. When the value of $\beta$ is positive, the auxiliary variable $a$ works as positive feedback to the neural activity $x$. (Note that these values should satisfy $\gamma > \beta$ because $x$ will diverge otherwise.)

When the decay rate $\gamma$ is large, i.e., the characteristic time scale of $a$ is small, the single-cell dynamics is effectively described only by the variable $x$ because the external input does not largely increase the value of $a$. It immediately vanishes with the termination of the input. Thus, the model behaves as a normal neuron without graded persistency, whose activity promptly decays without the external input (Fig 1, left panels). On the contrary, if the decay rate is small (Fig 1, right panels), the value of $a$ is almost kept constant even without external inputs. Due to the finite support of the auxiliary variable, the neural activity can be sustained, avoiding decay to zero, even without external input, which quantitatively reproduces the graded-persistent activity. The sustained activity gradually increases or decreases, reflecting the input history of the neuron, which agrees with experimental findings [11].

**Heterogeneous network with a subset of the GPA neurons.** Using the single neuron model, define the heterogeneous network whose subset neurons have the graded persistency

$$\dot{x}_i(t) = -x_i(t) + a_i(t) + \sum_{j=1}^{N} J_{ij}\phi(x_j(t)) + I_i(t)$$

$$\dot{a}_i(t) = -\gamma_i a_i(t) + \beta_i x_i(t).$$

(2)

Here, $i = 1, \dots, N$ denotes the index of neurons, $N$ is the number of neurons in the network, and $J = (J_{ij})$ is the connection weight matrix where $J_{ij}$ represents the synaptic connection strength from the $j$th to the $i$th neuron. To avoid self-connection, we set the diagonal components of $J$ as $J_{ii} = 0$ for all $i$. The values of the off-diagonal components $J_{ij}$ are independently chosen from a Gaussian distribution with variance $g^2/N$, i.e., $J_{ij} \sim \mathcal{N}(0, g^2/N)$. Here, the parameter $g$ controls coupling strength. In all numerical simulations in this paper, we use

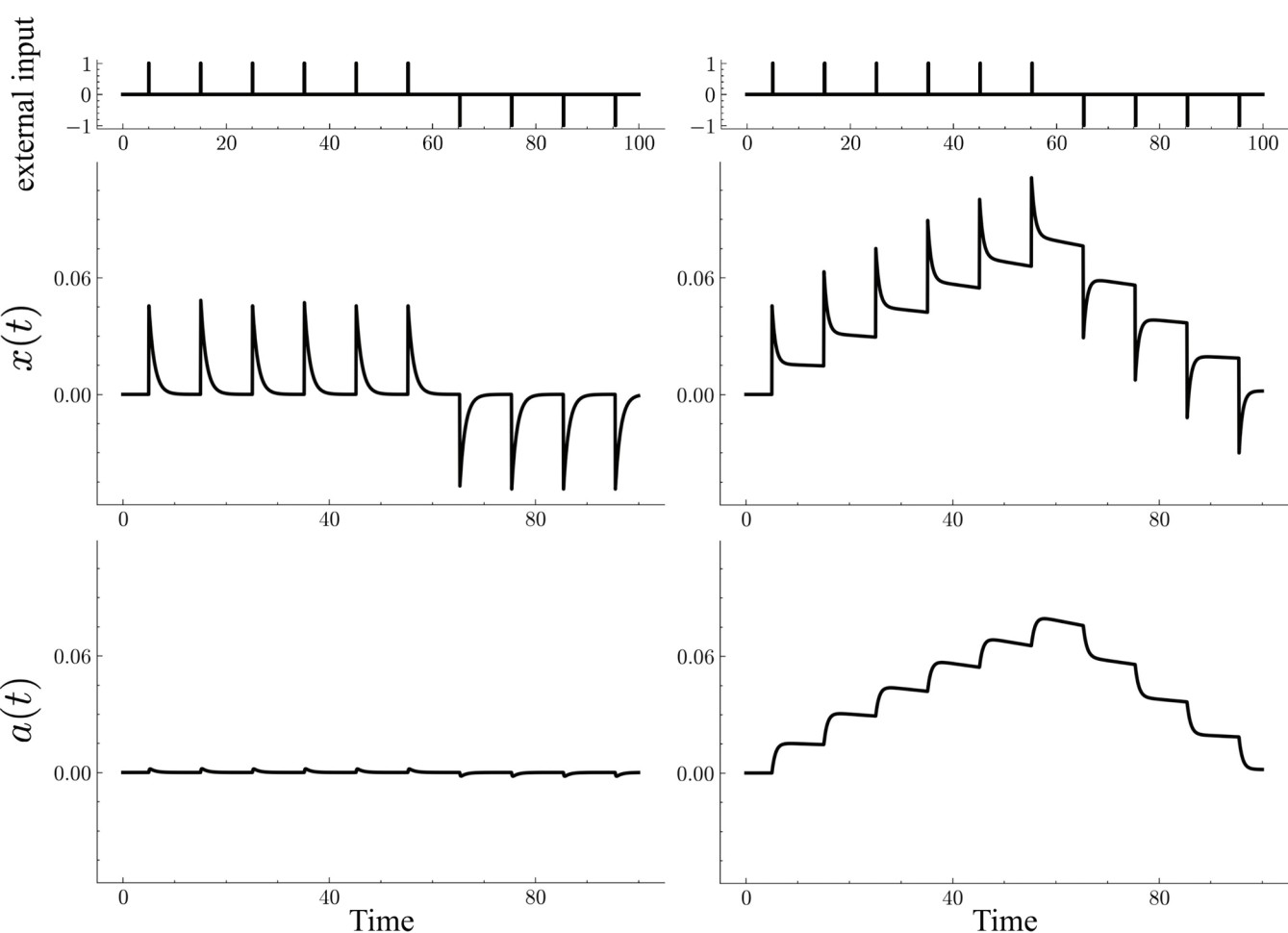

**Fig 1. Typical dynamics of the model neuron, Eq (1), with and without the graded-persistent activity (GPA) when successive pulse external stimulus (top panels) are given.** For a normal neuron (left panels, $\beta = 0.5, \gamma = 10$), the cell activity (left middle panel) decays rapidly when the pulse input is terminated as its auxiliary variable (left bottom panel) is kept small. By contrast, for a GPA neuron (right panels, $\beta = 0.5, \gamma = 0.51$), the cell activity (right middle panel) is kept almost constant, even during intervals between pulse external inputs. The sustained cell activity gradually increases in response to repeated depolarizing pulse inputs and decreases in response to subsequent hyperpolarizing pulse inputs. These sustained activities are supported by the slow decay of the auxiliary variable (right bottom panel).

$\phi(x) = \tanh(x)$, while the results of the theoretical analysis are not restricted to this particular choice of the activation function.

To make the neurons in the network heterogeneous, we randomly and independently chose the decay rates $\gamma_i$ from a two-point distribution:

$$\gamma_i = \begin{cases} \gamma_{low} & \text{probability } p \\ \gamma_{high} & \text{probability } 1 - p, \end{cases} \tag{3}$$

where $\gamma_{low} < \gamma_{high}$. Because a normal neuron is modeled by a high value of the decay rate, the parameter $p$, determining the ratio of the neurons with the low value of $\gamma$, controls the ratio of GPA neurons in the network. For the feedback strength parameter $\beta_i$, we use a uniform value $\beta_i = 0.5$ unless stated otherwise. (Note that, while we use a two-valued distribution for

simplicity, the theory developed here is not limited to discrete parameter distributions. For details, please see the Methods section, S1 Appendix, S1 and S2 Figs.)

Fig 2 shows typical temporal dynamics of the network for values of model parameters. The increase in coupling strength $g$ moves the network from a quiescent state (from the left to the right panels in Fig 2), where the activities promptly decay to zero, to a seemingly chaotic state where irregular neural activities are sustained. This result agrees with previous studies that reported the existence of the chaos-order transition at $g = g_c$, where $g_c$ is the transition point [21,28].

When the ratio $p$ of graded-persistent neurons in the network is increased (from the top to the bottom panels in Fig 2), the transition from the quiescent state to the chaotic state occurs at smaller values of $g$. This result suggests that the transition point itself is shifted toward expanding the chaotic regime by introducing GPA neurons to the network. The observation is of particular interest because the results of numerical simulations show that graded persistent neurons and other normal neurons behave almost similarly in the network (Fig 3). In the next section, we will see that this is actually the case and derive an equation determining $g_c$ as a function of the model parameters, including the ratio $p$.

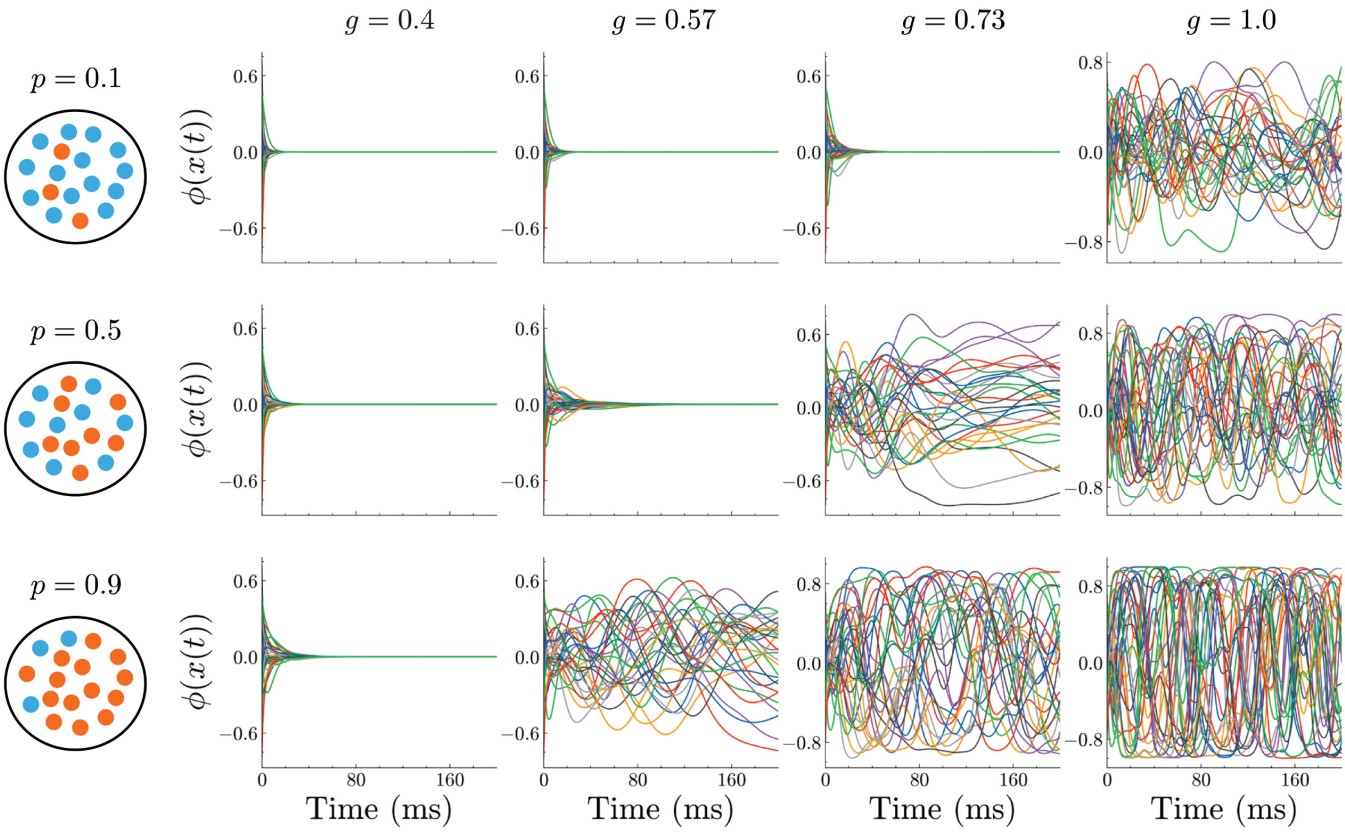

**Fig 2. Typical dynamics of neurons in the random network, Eq (2), for values of the coupling strength $g$ and the ratio of the GPA neurons $p$.** Panels are arranged such that the coupling strength $g$ increases from left to right ($g = 0.4, 0.57, 0.73, 1$), and the ratio of the GPA neurons increases from top to bottom ($p = 0.1, 0.5, 0.9$). For the larger value of the ratio of GPA, $p$, the transition from the silent to the chaotic state occurs at the smaller value of $g$. Other parameters are $N = 3000$, $\gamma_{high} = 10$, $\gamma_{low} = 1$, and $\beta = 0.5$.

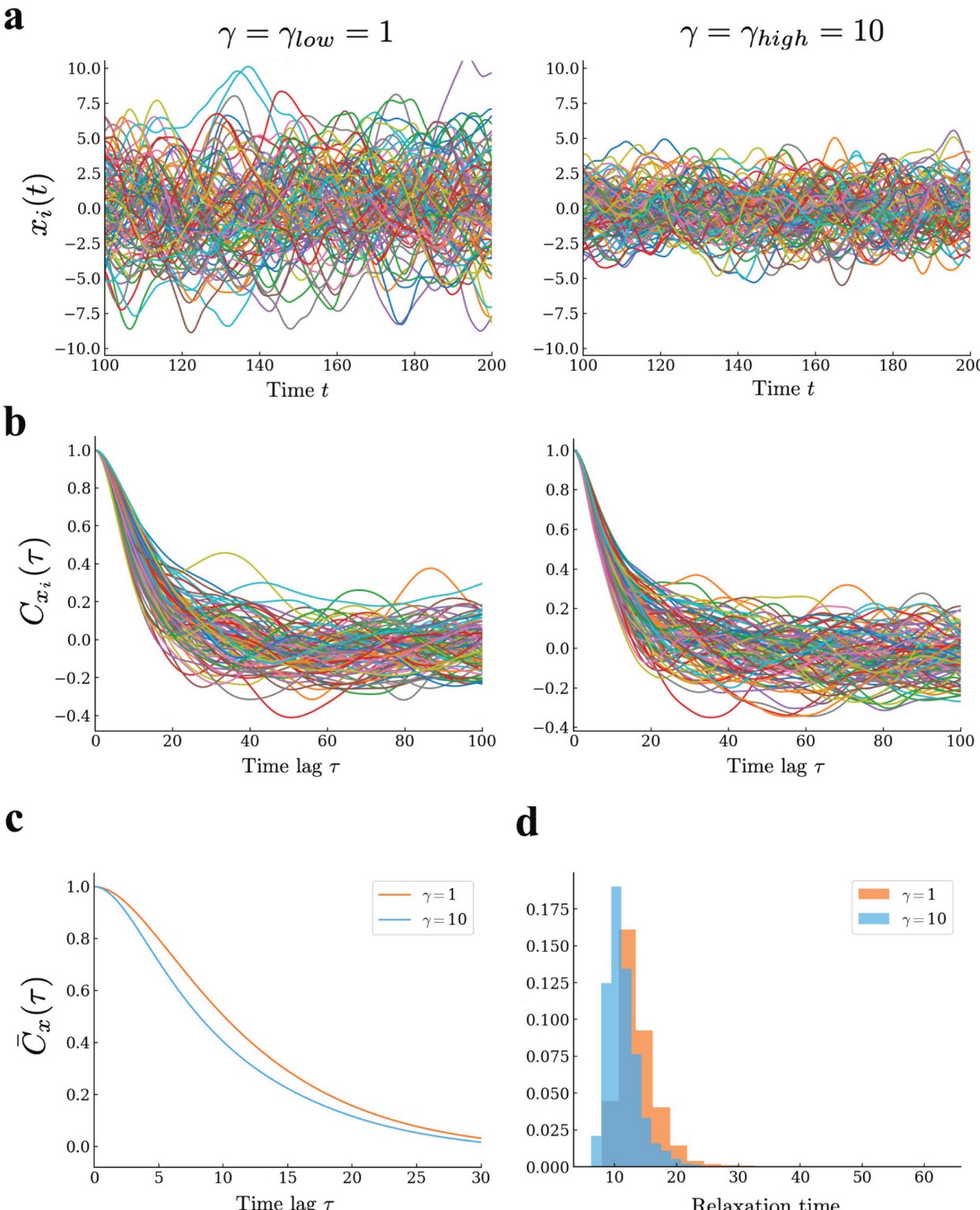

**Fig 3. Behavior of graded persistent neurons and other normal neurons in a network, Eq (2), operating in the chaotic regime.** (a) Temporal profiles of the activity of neurons in each population. (b) Autocorrelation functions of the time series for individual neurons. (c) Autocorrelation functions averaged over neurons in each population. (d) Density distributions of the correlation times of neurons in each population, obtained directly from the autocorrelation functions in (b). Parameters $N = 5000$ $p = 0.5$ and $g = 2.0$ are used. All results indicate that graded persistent neurons and other normal neurons behave similarly in the network under chaotic dynamics.

### Dynamical Mean Field Theory for the network with highly heterogeneous neurons

To analyze the shift of the transition point induced by the intrinsic heterogeneity of neurons, we use Dynamical Mean Field Theory (DMFT) [19,21,22,24–33]. DMFT mitigates the difficulty of studying the population dynamics of high-dimensional nonlinear systems by allowing us to replace the interaction term in each neuron's dynamics, i.e., the third term of Eq (2), with an effective dynamical mean field. The properties of this mean-field are determined by the neural dynamics evolving under the mean field, leading to a self-consistent equation for the order parameters that characterize the macroscopic network dynamics.

In the conventional procedure of DMFT, one can obtain a single equation of motion driven by the mean-field to describe the dynamics of all neurons in the network due to their homogeneity. Unlike these conventional approaches, the heterogeneity of neurons cannot be averaged out from the network (See Method for and following paragraphs details). Instead, applying the DMFT framework leads to a set of $N$ differential equations governing neural dynamics, with the heterogeneity preserved. The difference between the two arises from the system size dependence of the terms in the generating functional of mean-field theory: connection heterogeneity appears as a sum of $N^2$ terms, while neuron heterogeneity appears as a sum of $N$ terms; thus, one cannot treat them the same way. However, we will see that these equations can be averaged in Fourier space, resulting in a single equation that characterizes the chaos-order transition of the heterogeneous network.

Following previous works [21,27,28], let us assume that the external input $I_i$ to each neuron is an independent realization of a Gaussian process with zero mean. Then, by adapting the path integral method to the dynamical equations, Eq (2), one can replace the interaction term in each neuron's dynamics with another Gaussian process $\eta_\phi(t)$ (See Method for details):

$$\begin{aligned}
\dot{x}_i(t) &= -x_i(t) + a_i(t) + \eta_\phi(t) + I_i(t) \\
\dot{a}_i(t) &= -\gamma_i a_i(t) + \beta_i x_i(t)
\end{aligned}, \tag{4}$$

where the mean of $\eta_\phi(t)$ is zero, and its autocorrelation should be determined self-consistently by the constraint:

$$\langle \eta_\phi(t)\eta_\phi(t') \rangle = \frac{g^2}{N} \sum_{i=1}^{N} \langle \phi(x_i(t))\phi(x_i(t')) \rangle. \tag{5}$$

If the network consisted of homogeneous neurons, the above equations would be the same for all neurons, and we could remove the subscript $i$ representing the neural index. However, we must keep the index because the decay rates $\gamma_i$ differ across neurons. Note that a previous study proposed eliminating similar intrinsic heterogeneity by averaging the above equations over all neurons [27]. However, we will demonstrate in the next section that such a naive averaging fails to deliver accurate results.

Whereas we cannot reduce the set of equations Eq (4) to a single one, it is still possible to obtain a single equation determining the transition point of the network from the set of equations (See Method for details). The Fourier transform of the above equations gives

$$\begin{cases}
i\omega X_i^L(\omega) &= -X_i^L(\omega) + A_i^L(\omega) + H_\phi^L(\omega) \\
i\omega A_i^L(\omega) &= -\gamma_i A_i^L(\omega) + \beta X_i^L(\omega)
\end{cases}, \tag{6}$$

where $X_i^L$, $A_i^L$, and $H_\phi^L$ are the normalized Fourier transforms of $x_i$, $a_i$, and $\eta_\phi$, respectively, that is, $X_i^L(\omega) = \mathcal{F}^L[x_i(t)](\omega)$, $A_i^L(\omega) = \mathcal{F}^L[a_i(t)](\omega)$, and $H_\phi^L(\omega) = \mathcal{F}^L[\eta_\phi(t)](\omega)$, where $\mathcal{F}^L[x(t)](\omega) = \frac{1}{\sqrt{L}} \int_{-L/2}^{L/2} x(t) \exp(-i\omega t)dt$. Eliminating $A_i$ from the above and multiplying the resulting equation by its complex conjugate, we have

$$S_{x_i}(\omega) = g^2 G(\omega; \gamma_i, \beta)\bar{S}_\phi(\omega) \tag{7}$$

$$G(\omega; \gamma, \beta) = \frac{\omega^2 + \gamma^2}{\omega^4 + (\gamma^2 + 2\beta + 1)\omega^2 + (\gamma - \beta)^2}, \tag{8}$$

where $S_{x_j}(\omega) = \mathcal{F}\left[\langle x_j(t+\tau)x_j(t)\rangle\right]$ is the power spectral density of the activity of the $i$th neuron $x_i$, $\bar{S}_\phi(\omega) = \frac{1}{N}\sum_{i=1}^N \mathcal{F}\left[\langle \phi(x_i(t))\phi(x_i(t'))\rangle\right]$ is the average power spectral density of the activity via activation function $\phi(x)$. One can safely average Eq (7) over all neurons because, unlike Eqs (2) and (4), it does not have higher-order terms of heterogeneity, such as $\gamma_i a_i$, which would result in additional unknown terms. The average gives

$$\bar{S}_x(\omega) := \frac{1}{N}\sum_{i=1}^N S_{x_i}(\omega) \tag{9}$$

$$= g^2 \bar{S}_\phi(\omega) \frac{1}{N}\sum_{i=1}^N G(\omega; \gamma_i, \beta) \tag{10}$$

$$\rightarrow g^2 \bar{S}_\phi(\omega)\langle G(\omega; \gamma, \beta)\rangle_{\gamma,\beta} = g^2 \bar{S}_\phi(\omega)\bar{G}(\omega), \tag{11}$$

where in the third line, we have taken the limit of $N \rightarrow \infty$ and used the law of large numbers, defining $\bar{G}(\omega) := \langle G(\omega; \gamma, \beta)\rangle_{\gamma,\beta}$.

Now, let us assume that the activation function $\phi(x)$ satisfies the condition

$$|\phi(x)| \leq |x|, \tag{12}$$

which is certainly satisfied by most of the generally used activation functions, including $\tanh(x)$ we used. This condition leads to an inequality between the power spectral densities:

$$0 \leq \int_{-\infty}^{\infty} \bar{S}_\phi(\omega)\, d\omega \leq \int_{-\infty}^{\infty} \bar{S}_x(\omega)\, d\omega, \tag{13}$$

which, when combining with equation Eq (11), gives the required equation determining the transition point $g_c$ of the heterogeneous network:

$$\max_\omega \bar{G}(\omega)g_c^2 = 1. \tag{14}$$

## Introduction of GPA neurons expands the dynamical regime of the network

When the decay rate $\gamma_i$ of each neuron is independently chosen from the two-point distribution, Eq (3), one can obtain an analytical expression of $\bar{G}(\omega)$ and the transition point $g_c$ explicitly:

$$g_c(p, \gamma_{low}, \gamma_{high}, \beta) = \left( p\left(\frac{\gamma_{low}}{\gamma_{low} - \beta}\right)^2 + (1-p)\left(\frac{\gamma_{high}}{\gamma_{high} - \beta}\right)^2 \right)^{-\frac{1}{2}}. \tag{15}$$

If the network consists of homogeneous neurons (i.e., $p = 0$), this expression simplifies to $g_c = 1 - \beta/\gamma_{high}$, which reproduces the conventional result $g_c = 1$ in the limit as infinite decay rate ($\gamma_{high} \to \infty$) or no feedback ($\beta \to 0$), where the single neuron model reduces to a conventional one-dimensional dynamical system without the auxiliary slow variable.

If one increases the heterogeneity of the network, the transition point $g_c$ generally decreases from one, as $g_c$ is a monotonically decreasing function of $p$ since $\gamma_{low}/(\gamma_{low} - \beta) > \gamma_{high}/(\gamma_{high} - \beta)$. Therefore, introducing GPA neurons with slow dynamics to the network consistently shifts the network toward a more dynamic regime, i.e., a state preferable for temporal information encoding.

To validate the theoretical prediction and to examine how the shift of the transition point is affected by model parameters, we performed numerical simulations of the network dynamics in Eq (2) and compared them with the theoretical prediction Eq (15) for various values of the model parameters. To numerically identify the transition point, we calculated the maximum value of the power spectrum $\max_\omega \bar{S}_x(\omega) = \max_\omega 1/N \sum_{i=1}^{N} S_{x_i}(\omega)$ from the numerically obtained time series. Since the power spectrum is nonnegative by definition, a positive maximum indicates that the network is in a dynamic regime. In contrast, the zero maximum implies the network has converged to a steady state.

Fig 4 shows the results. We can see that the theoretical predictions (red lines) agree well with the results of numerical simulations in all cases, and the dynamical regime consistently expands as the heterogeneity $p$ increases from 0. This expansion of the dynamical regime is particularly pronounced when the decay rate of GPA neurons $\gamma_{low}$ is small (the far left panel of Fig 4a) and the feedback strength $\beta$ is large (the far right panel of Fig 4c), where GPA neurons exhibit slower auxiliary dynamics (small $\gamma_{low}$) and have a greater influence on the neural activity $x$ (large $\beta$). This is because the slow auxiliary variable of GPA neurons facilitates the nonzero firing activity of other neurons in the network and helps to prevent the population dynamics from converging to a quiescent state.

## Analysis of the transition induced by the introduction of GPA neurons

To more precisely examine the transition of the network from the quiescent state to the dynamical state, we numerically calculated the eigenspectrum (Fig 5) and the maximum Lyapunov exponent (Fig 6) of the network for values of the ratio of GPA neurons $p$ and the coupling strength $g$.

The eigenspectrum (Fig 5) of a fixed point in the network characterizes the system's linear stability around that point. For recurrent neural networks with random connections, the instability of the trivial fixed point often corresponds to the chaos-order transition point, as the loss of this stability may lead to chaotic behavior due to the network's intrinsic randomness [21]. We linearized the model system, Eq (2), around its trivial fixed point, or the origin, and numerically computed the eigenspectrum of the Jacobian matrix. The panels in Fig 5 are arranged such that the coupling strength, $g$, increases from top to bottom, and the ratio of the GPA neuron, $p$, increases from left to right. The trivial fixed point is stable if the real parts of all eigenvalues are negative and unstable if at least one has a positive real part. The eigenvalue distributions exhibit complex, nontrivial shapes on the complex plane. However, it is evident that instability, indicated by the appearance of a positive eigenvalue, occurs at smaller values of $g$ for larger values of $p$, as predicted by the theoretical results.

The maximum Lyapunov exponent is used to characterize the complexity of nonlinear systems, particularly their chaotic behavior, as a positive exponent indicates the network's

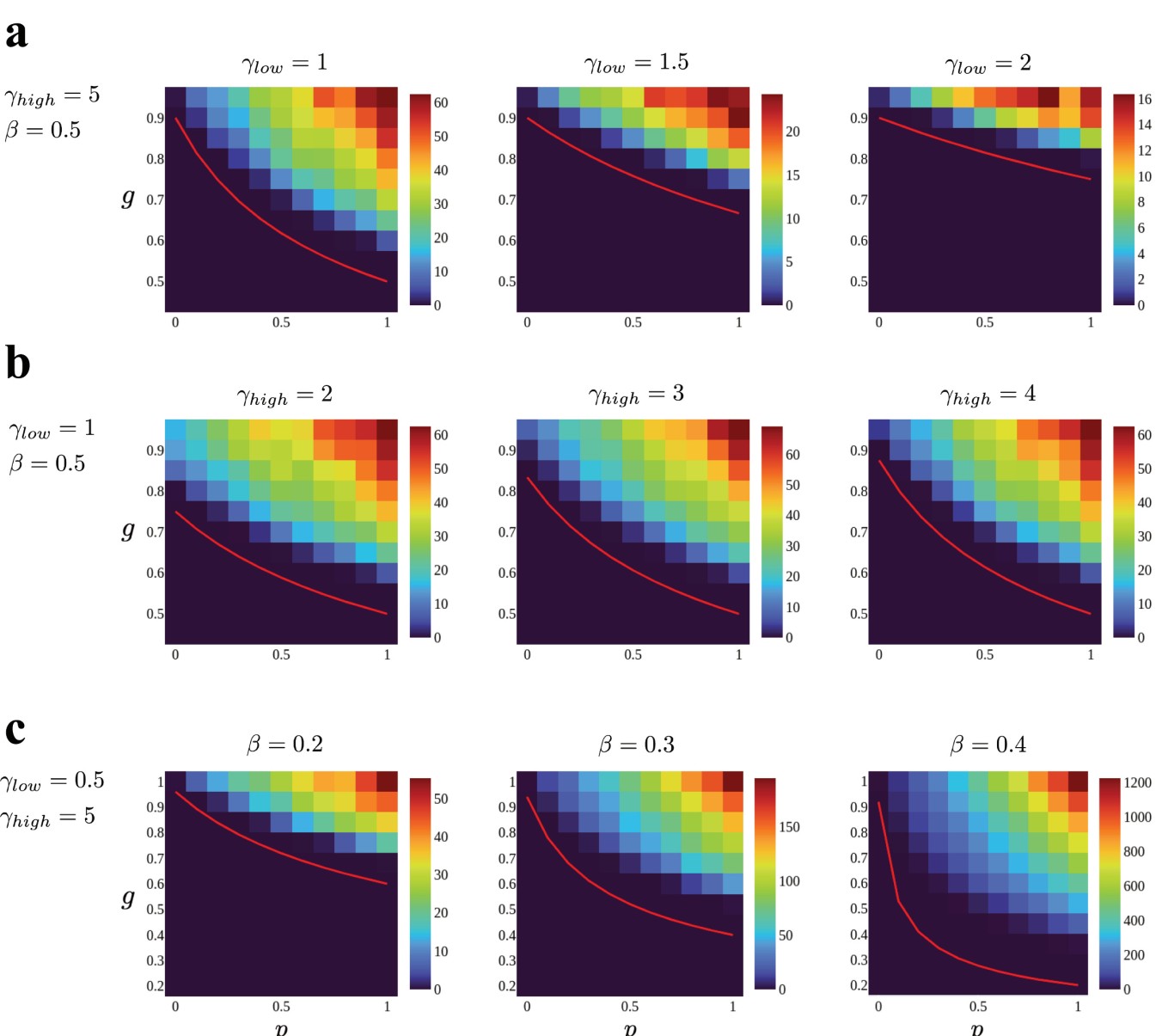

**Fig 4. Maximum power spectrum of the network dynamics for values of the ratio of the GPA neuron $p$ (horizontal axis of each panel) and coupling strength $g$ (vertical axis of each panel).** Red curves are the theoretical prediction of the transition point, $g_c$, given by Eq (15). **a:** $\gamma_{high} = 5$ and $\beta = 0.5$ are fixed and $\gamma_{low}$ increases from left to right ($\gamma_{low} = 1, 1.5, 2.0$). **b:** $\gamma_{low} = 1$ and $\beta = 0.5$ are fixed and $\gamma_{high}$ increases from left to right ($\gamma_{high} = 2, 3, 4$). **c:** $\gamma_{low} = 0.5$ and $\gamma_{high} = 5$ are fixed and $\beta$ increases from left to right ($\beta = 0.2, 0.3, 0.4$). Numerical simulations are performed with $N = 3000$ for all panels.

chaotic state. To compute the maximum Lyapunov exponent, we employed the method proposed by Benettin et al. [40]. This involves numerically solving the model equations, Eq (2), to obtain the trajectory and estimation of the exponent from the growth rate of the tangent vector along the trajectory. Fig 6a shows that the maximum Lyapunov exponent changes sign from negative to positive at points close to the theoretical predictions. While small discrepancies exist between the numerical simulations and theoretical results, we confirmed that these mismatches arise from finite-size effects. As the number of neurons in the network increases,

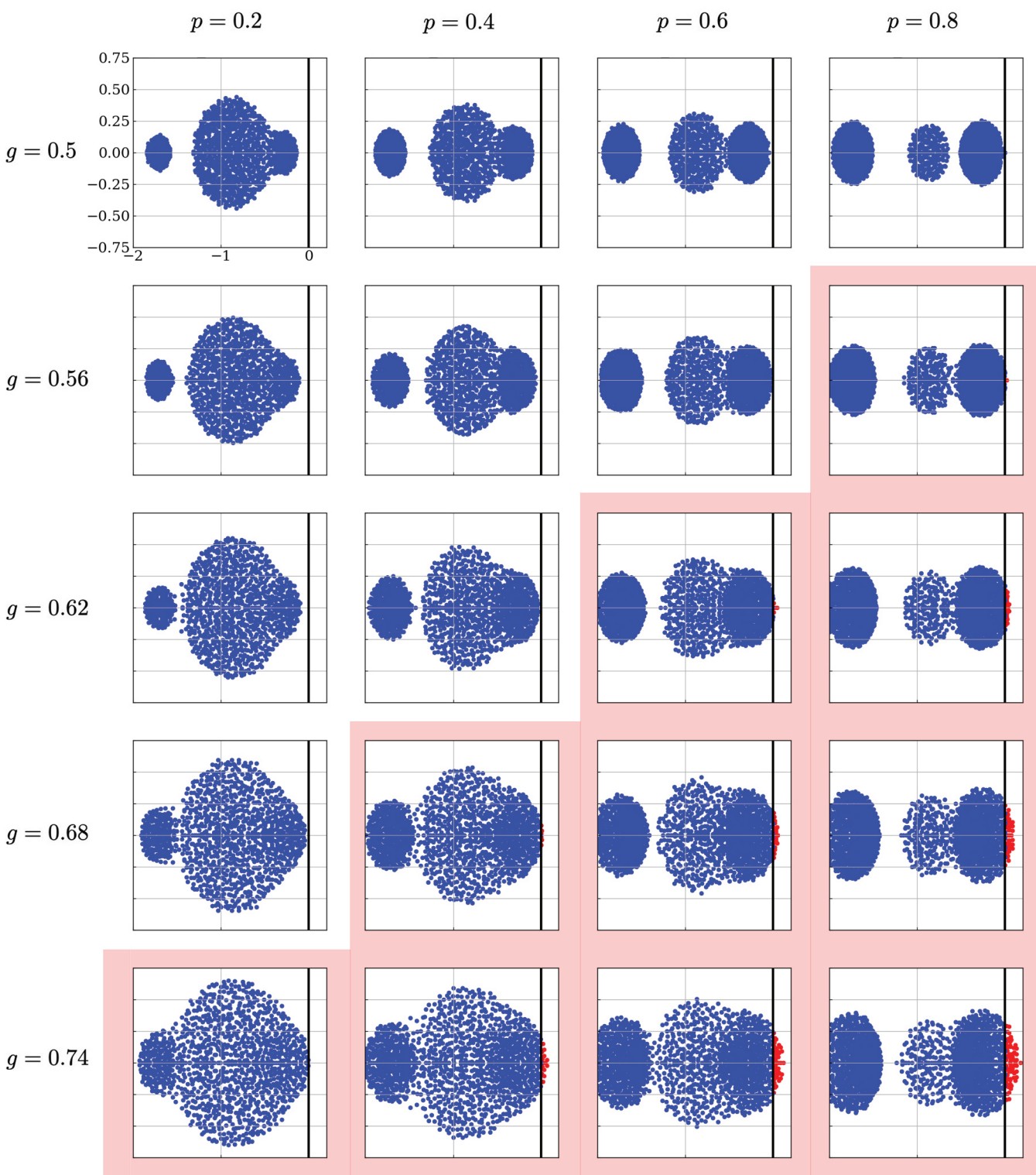

**Fig 5. Eigen spectrum in the complex plane for the Jacobian at the trivial fixed point of the network dynamics.** Dots in each panel indicate the eigenvalue of the linear stability matrix on the complex plane. Blue and red dots indicate eigenvalues with negative and positive real parts, respectively. Panels are arranged such that the GPA neuron ratio, $p$, increases from left to right, and coupling strength, $g$, increases from top to bottom. The thick vertical line in each panel is the imaginary axis. Panels shaded by red background mean the coupling strength of the network is above the transition point predicted theoretically, i.e., $g > g_c$. Other model parameters are $N = 1000$, $\gamma_{low} = 1$, $\gamma_{high} = 5$ and $\beta = 0.5$.

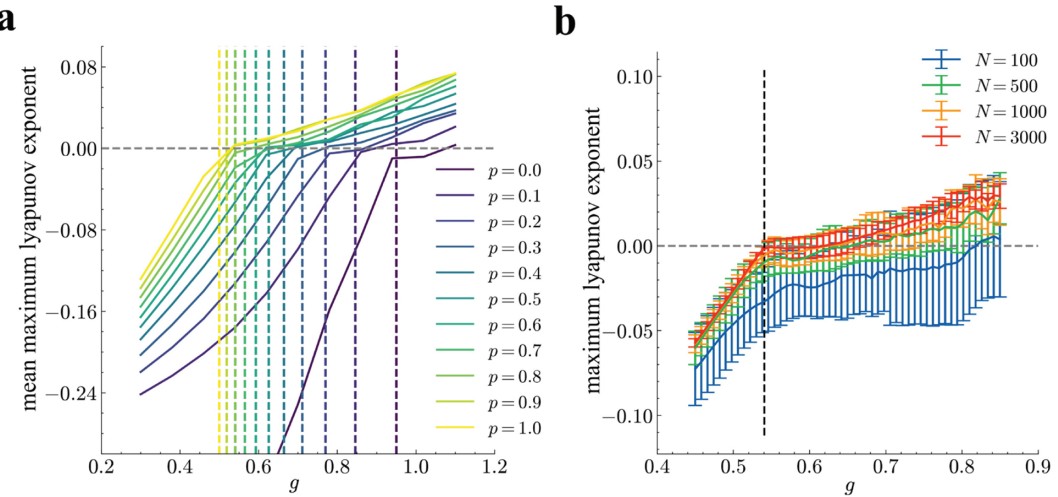

**Fig 6. The maximum Lyapunov exponent of the network dynamics. a**: The numerically estimated maximum Lyapunov exponent, averaged over ten realizations of random initial conditions, for the model network as a function of the coupling strength $g$. Line colors indicate the value of $p$ (from the far-right purple curve for $p = 0.0$ to the far-left yellow curve for $p = 1.0$). Dashed vertical lines indicate $g_c$. The number of neurons in the network is fixed at $N = 3000$. **b**: The same Lyapunov exponent for different values of $N$ with $p = 0.8$. Blue, orange, green, and red lines correspond to networks of $N = 100, 500, 1000,$ and $3000$, respectively. Error bars show standard deviation.

the discrepancies decrease (Fig 6b), which is consistent with the fact that DMFT assumes an infinite number of neurons.

## Gaussian heterogeneity of adaptation of neurons in random networks

To demonstrate the applicability of the developed theory, we apply it to a different type of neural heterogeneity: Gaussian-distributed adaptation of each neuron in the network. In a pioneering theoretical work studying how intrinsic properties of single neurons modulate population dynamics in random neural networks, Muscinelli et al. proposed a two-dimensional neuron model with strong adaptation [27]. Remarkably, this adaptation neuron model is equivalent to the GPA neuron model introduced here but with the opposite sign of $a$ in the equation for neural activity. Specifically, the sign of the second term in Eqs (1) and (2) is negative, rather than positive, in the adaptation neuron model. Due to the equivalence, it is straightforward to apply our method to a heterogeneous network of the adaptation neuron model.

In the previous study, the authors introduced heterogeneity in adaptation for each neuron by randomly choosing the feedback strength parameter $\beta_i$ from a Gaussian distribution with mean $\mu_\beta < 0$ and small variance $\sigma_\beta^2$, i.e., $\beta_i \sim \mathcal{N}(\mu_\beta, \sigma_\beta^2)$, using the variance to characterize the heterogeneity. (The variance was kept small to ensure that the sampled values of $\beta_i$ remained negative.) Unlike the theory developed here, the authors treated the heterogeneity similarly to the coupling strengths between neurons. They naively assumed that the heterogeneity could be effectively expressed by an additional Gaussian mean field in the DMFT. Based on this assumption and following the conventional DMFT procedure, they derived a single mean-field equation driven by two Gaussian processes, which gives the equation determining the transition point $\hat{g}_c$ of the heterogeneous network:

$$\max_{\omega} G_{\beta}(\omega)\hat{g}_c^2 = 1 \qquad (16)$$

$$G_{\beta}(\omega) = \frac{G(\omega;\gamma,\mu_{\beta}<0)}{1 - \frac{\sigma_{\beta}^2}{\gamma^2+\omega^2} G(\omega;\gamma,\mu_{\beta}<0)}, \qquad (17)$$

where the function $G$ is given by Eq (8). To emphasize the difference between this transition point and the one obtained in the next section, we denoted the transition point obtained here with hat as $\hat{g}_c$.

However, the assumption of the previous study that the intrinsic properties of neurons can be expressed by an additional Gaussian process was not entirely correct. (Technically speaking, averaging the generating functional of the model equation is still possible even with this type of heterogeneity. However, the equation cannot be reduced to a single expression. This is because the system-size dependence of the term arising from heterogeneity scales differently from that of the connection heterogeneity (see Method for details)). Instead, we need to deal with a set of $N$ differential equations, which must be averaged in Fourier space to determine the transition point of the network, as described in the previous section. Following the same procedure as in the previous section, we have the equation for the transition point:

$$\max_{\omega} \langle G(\omega;\gamma,\beta)\rangle_{\beta}\, g_c^2 = 1, \qquad (18)$$

where the function $G(\omega;\gamma,\beta)$ is given by Eq (8) and $\langle G(\omega;\gamma,\beta)\rangle_{\beta}$ represents the average of the function over sampled values of $\beta_i$ in the network.

Interestingly, the equations for the transition point derived in the previous study and ours can predict opposite tendencies regarding how this heterogeneity shifts the chaos-order transition point in certain cases. The equation of the previous study predicts that the transition point decreases with increasing $\sigma_{\beta}$. Oppositely, the equation derived here predicts that the transition point will increase, meaning the dynamical regime will shrink due to this type of heterogeneity. To test these predictions, we numerically simulated the population dynamics for various model parameters. Fig 7 shows that, as predicted by the current theory, the transition point increases as $\sigma_{\beta}$ increases, whereas $\hat{g}_c$ shows the opposite trend. Further exploration of the model parameters causing this discrepancy between the two theories is an important future subject.

## Discussion

In this study, considering the properties of the entorhinal cortex, we develop a theory to describe the population dynamics of random neural networks consisting of highly heterogeneous neurons. We propose a simple two-dimensional neuron model representing both normal neurons and neurons exhibiting graded persistent activity (GPA). We extend the well-established theoretical tool, DMFT, which provides an effective mean-field equation for network dynamics, to cases where the neurons in the network are highly heterogeneous. Unlike conventional DMFT, the derived mean-field model consists of $N$-dimensional stochastic equations rather than a single equation. However, we showed that averaging these equations is still possible in Fourier space. This allows us to theoretically determine the transition point of the network by focusing on the average power spectrum of the network dynamics. Since recent experiments have revealed that cortical neurons are highly heterogeneous in their intrinsic properties, the theory developed here will be an important tool, alongside

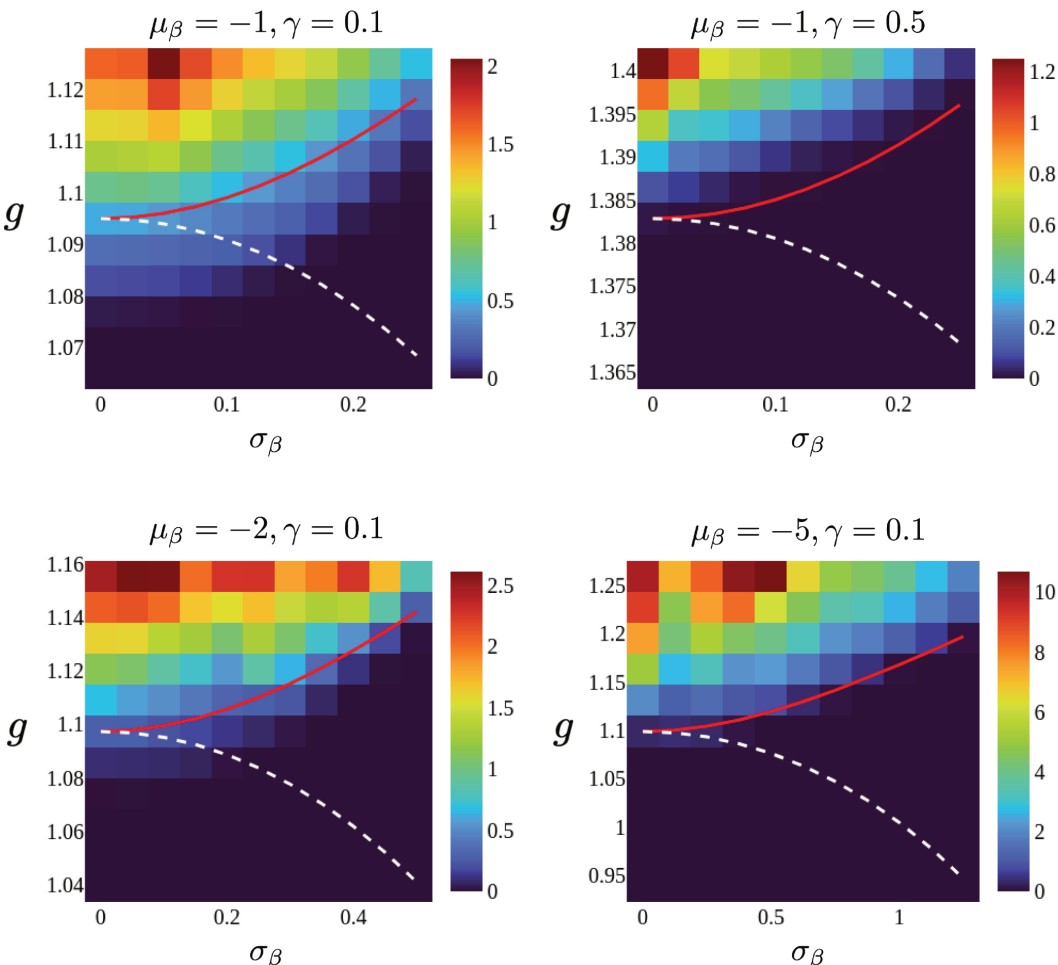

**Fig 7. Maximum power spectrum of the network dynamics with Gaussian heterogeneous adaptation for values of model parameters.** The horizontal and vertical axes are $\sigma_\beta$ and $g$, respectively. The red curves show theoretical prediction $g_c$ obtained using the method developed in this study The white dashed lines indicate $\hat{g}_c$ that are derived from a naïve approximation of the heterogeneity.

conventional DMFT, for clarifying the properties of recurrent networks of realistic neuron models.

The results of the theory suggest that the stable region of the network shrinks, and the network becomes more dynamic as the ratio of GPA neurons increases, regardless of other model parameters. Since network activity triggered by external inputs does not decay near the boundary between the stable and dynamical regimes, lowering the coupling strengths required to reach this boundary may facilitate the encoding of long-timescale information in the entorhinal cortex. This boundary is widely known as the "edge of chaos" in the context of reservoir computing and is associated with optimal computational capabilities.

To validate this concept within our framework, we numerically calculated the memory capacity, a common metric in reservoir computing [34] (See S1 Appendix for details). As demonstrated S4 Fig, the memory capacity is indeed maximized at the onset of the chaotic regime, just below the critical transition point predicted by our theory. This finding not only supports the computational advantages of operating at the edge of chaos but also implies a

novel tuning mechanism: when the available range of synaptic coupling strengths is limited, a network may still be able to achieve its maximal memory capacity by modulating its intrinsic heterogeneity (See S1 Appendix for details).

Of particular interest is the relationship between our study and the computational model of temporal representation proposed by Shankar and Howard [41], which employs a set of leaky integrators with a spectrum of decay timescales to encode stimulus history. Our work and theirs address distinct, though related, questions. While our study investigates how the existence of neurons with diverse timescales impacts the collective dynamics of a network, Shankar and Howard aimed to answer the more specific computational question of how these diverse timescales can be utilized to construct a scale-invariant representation of past events. This divergence in research goals is reflected in their respective methodologies. In contrast to our use of an extended DMFT to analyze a random network's statistical properties, Shankar and Howard proposed a specifically structured, layered network architecture designed to approximate an inverse Laplace transform of the stimulus history. Consequently, the functional role of heterogeneity is also interpreted differently. In our framework, the diversity of neural timescales acts as a modulator that alters the network's global dynamical regime, whereas in their model, this same diversity is the essential substrate for building the temporal representation itself. Despite these differences, the two frameworks are not contradictory but are better viewed as complementary. The population of leaky integrators posited by Shankar and Howard could, for instance, be biologically realized by the heterogeneous neural population that includes GPA neurons, as modeled in our study. An integration of these two perspectives could therefore provide a more comprehensive understanding that bridges the gap from the role of diversity in single elements and population dynamics to the emergence of higher-order cognitive functions like temporal representation.

The networks considered here belong to a class of systems that maintain memory of past inputs through transient trajectories, rather than storing information in stable attractors like traditional Hopfield networks. The capacity for such temporal coding is thought to depend critically on non-normal dynamics, which can transiently amplify specific inputs [42,43]. Our model, based on random recurrent interactions, naturally gives rise to this non-normal property. This raises the intriguing possibility that by tuning the nature of the heterogeneity, for instance, by altering the specific variables or the shape of their statistical distributions, our framework could account for specific, dynamic coding phenomena. A compelling target for such investigation would be the transient amplification of odor representations observed experimentally in the insect antennal lobe, a phenomenon that has also been linked to non-normal network dynamics [44,45].

Experimental findings suggest that characteristic neural dynamics encoding time arises from the interaction between the network's internal dynamics and external signals to the network. In general, when external signals are applied to a nonlinear system, they tend to entrain the system and increase its stability [22,46–54]. Therefore, if the internally generated intrinsic dynamics of the network are in a stable and quiescent state, the system struggles to integrate external signals. This is because the signals further stabilize the network, causing input information to decay too quickly to be retained.

In contrast, if the network operates in a dynamic regime near the transition point, external signals can entrain the network and drive its dynamics closer to or just below the transition point. This is the ideal state for integrating external inputs and maintaining their information over a long time. Thus, an important direction for future research will be to extend the theory to networks influenced by external signals. Additionally, it is also an important future task to investigate the relationship between the network's intrinsic dynamics and the characteristic population activity of the entorhinal cortex, such as the activity of time cells

and the ramping activity. In this regard, we have demonstrated that neurons in our heterogeneous recurrent network can reproduce response patterns similar to those of 'temporal context cells' recently observed in the entorhinal cortex, when intrinsic time scales, in particular the feedback strengths $\beta_i$, follow continuous distributions (see S1 Appendix and S3 Fig for details).

We also applied the theory to a different type of heterogeneity studied in previous work: neuronal adaptation. The theory developed here accurately describes the numerical results, demonstrating that heterogeneity in adaptation can reduce the dynamical regime of the network. However, we show that the predictions of the previous study do not always capture the direction of the shift in the transition point. This result highlights that intrinsic heterogeneity should be treated carefully and differently from the heterogeneity of coupling strengths. Another important message from this result comes from the comparison between Figs 4 and 7. Although both results show that introducing heterogeneity to the intrinsic properties of neurons modulates population dynamics and shifts the transition point, the directions of the shifts are entirely opposite, indicating that one cannot generally conclude whether heterogeneity expands or shrinks the dynamical regime of the network without knowing the specific details of the heterogeneity.

The chaos-order transition studied here has been recognized as playing an important role in temporal information processing, especially within the framework known as reservoir computing, where input sequences are stored in randomly connected recurrent networks. For instance, several studies have shown that the memory capacity of a neural population for input sequences is maximized near the transition point [55]. Therefore, an interesting future direction would be to evaluate the memory capacity of heterogeneous networks of GPA neurons by extending the theory developed here.

This study is closely related to recent important work by Stern et al. [19]. In their study, Stern et al. demonstrated that the heterogeneous distribution of cell assembly sizes can explain the experimentally observed heterogeneous time scales in neuronal circuits. By extending the DMFT, they showed that the heterogeneity of time scales gives rise to a novel chaotic regime characterized by bistable activity. To reveal this chaotic regime, their analysis employed random matrix theory, which can provide the eigenspectrum distribution of large random matrices [56]. In contrast, our approach demonstrates that the transition point of a heterogeneous network can be explicitly determined by a single equation, Eq (14), derived by directly averaging the heterogeneity in Fourier space. Given the difficulty in deriving the eigenspectrum distribution for most matrix ensembles, our results complement their work by offering an alternative framework to elucidate the significant role of intrinsic heterogeneity in large-scale systems. Further studies of the transition mechanism could deepen the connections between their theoretical advancements and ours.

Related to the above, another promising theoretical direction is the study of the eigenspectrum distribution of the model. As shown in Fig 5, the eigenvalues of the model network exhibit a highly intricate yet well-organized nontrivial structure. Several bulks or clusters of eigenvalues are observed, which gradually merge as the model parameters vary. Since the connection matrix of the network is randomly generated, such eigenspectrum distributions may be analyzed within the framework of random matrix theory [31,56–61] . Given the critical role of heterogeneity in a wide range of applications, understanding the mechanisms underlying the modulation of the eigenspectrum could provide deeper insights into the dynamical behavior of large, generally heterogeneous systems.

## Methods

### Dynamical mean field theory for the neural networks with heterogeneous parameters

In this section, within the framework of dynamical mean-field theory (DMFT) [19,21,22,24–33], we derive effective dynamics (Eq (4) in the main text) of each neuron from the original dynamics, i.e., Eq (2) in the main text:

$$\dot{x}_i(t) = -x_i(t) + a_i(t) + \sum_{j=1}^{N} J_{ij}\phi(x_j(t)) + I_i(t)$$

$$\dot{a}_i(t) = -\gamma_i a_i(t) + \beta_i x_i(t). \tag{19}$$

By formally adding white Gaussian noise $\xi_{m,i}^x$, $\xi_{m,i}^a$ to each term in the above equations and defining the right-hand sides as $C_i^x(t) := -x_i(t) + a_i(t) + \sum_{j=1}^{N} J_{ij}\phi(x_j(t)) + I_i(t)$ and $C_i^a(t) := -\gamma_i a_i(t) + \beta_i x_i(t)$, we obtain a set of stochastic equations:

$$\dot{x}_i(t) = C_i^x(t) + \xi_i^x(t)$$

$$\dot{a}_i(t) = C_i^a(t) + \xi_i^a(t). \tag{20}$$

Considering this as the Ito stochastic equation and discretizing it in time with interval $\Delta t$ yields

$$x_{m,i} - x_{m-1,i} - C_{m-1,i}^x \Delta t = \xi_{m,i}^x$$

$$a_{m,i} - a_{m-1,i} - C_{m-1,i}^a \Delta t = \xi_{m,i}^a, \tag{21}$$

where $x_{m,i} = x_i(m\Delta t)$, $a_{m,i} = a_i(m\Delta t)$, $C_{m,i}^x = C_i^x(m\Delta t)$, and $C_{m,i}^a = C_i^a(m\Delta t)$. Since Eq (21) give two-point relationships of variables, we can obtain the high-dimensional joint probability density function of $\{x_{m,i}, a_{m,i}\}_{m,i}$ marginalized over $\{\xi_{m,i}^x, \xi_{m,i}^a\}$ from this as

$$P(\{x_{m,i}, a_{m,i}\}_{m,i}) = \int \prod_{\alpha \in \{x,a\}} \prod_{i=1}^{N} \prod_{m=1}^{M} P(\xi_{m,i}^\alpha) P(\{x_{m,i}, a_{m,i}\}_{m,i} | \{\xi_{m,i}^x, \xi_{m,i}^a\}_{m,i}) d\xi_{m,i}^\alpha \tag{22}$$

$$= \prod_{\alpha \in \{x,a\}} \prod_{i=1}^{N} \prod_{m=1}^{M} \int P(\xi_{m,i}^\alpha) \delta(B_{m,i}^\alpha - \xi_{m,i}^\alpha) d\xi_{m,i}^\alpha$$

$$= \prod_{\alpha \in \{x,a\}} \prod_{i=1}^{N} \prod_{m=1}^{M} \int P(\xi_{m,i}^\alpha) \int_{-i\infty}^{i\infty} \exp(k_{m,i}^\alpha B_{m,i}^\alpha - k_{m,i}^\alpha \xi_{m,i}^\alpha) \frac{dk_{m,i}^\alpha}{2\pi i} d\xi_{m,i}^\alpha$$

$$= \prod_{\alpha \in \{x,a\}} \prod_{i=1}^{N} \prod_{m=1}^{M} \int_{-i\infty}^{i\infty} \exp(k_{m,i}^\alpha B_{m,i}^\alpha + \ln Z_\xi(-k_{m,i}^\alpha)) \frac{dk_{m,i}^\alpha}{2\pi i}, \tag{23}$$

where we have shortened the expression on the left-hand side of Eq (21) by defining $B_{m,i}^x = x_{m,i} - x_{m-1,i} - C_{m-1,i}^x \Delta t$ and $B_{m,i}^a = a_{m,i} - a_{m-1,i} - C_{m-1,i}^a \Delta t$. In the second line, we used the Fourier representation of the Dirac delta function, and in the third line, we introduced the moment generating function $Z_\xi(-k_{m,i}^\alpha)$ of the stochastic variable $\xi_{m,i}^\alpha$, defined by $Z_\xi(-k_{m,i}^\alpha) = \int P(\xi_{m,i}^\alpha) \exp(-k_{m,i}^\alpha \xi_{m,i}^\alpha) d\xi_{m,i}^\alpha$.

Let us consider the behavior of the stochastic distribution, Eq (23), in the limit of $\Delta t \to 0$, or $M \to \infty$. The first and the second terms of the exponential function converge as

$$\prod_{m=1}^{M} \exp(k_{m,i}^{\alpha} B_{m,i}^{\alpha}) = \exp\left(\sum_{m=1}^{M} k_{m,i}^{\alpha} \left(\frac{\alpha_{m,i} - \alpha_{m-1,i}}{\Delta t} - C_{m-1,i}^{\alpha}\right) \Delta t\right)$$

$$\xrightarrow{\Delta t \to 0, M \to \infty} \exp\left(\int k_i^{\alpha}(t)(\dot{\alpha}_i(t) - C_i^{\alpha}(t))dt\right) \tag{24}$$

and

$$\prod_{m=1}^{M} \exp\left(\ln Z_{\xi}(-k_{m,i}^{\alpha})\right) = \exp\left(\sum_{m=1}^{M} \left(k_{m,i}^{\alpha}\right)^2 \sigma_{\alpha}^2 \Delta t\right)$$

$$\xrightarrow{\Delta t \to 0, M \to \infty} \exp\left(\int \left(k_i^{\alpha}(t)\right)^2 \sigma_{\alpha}^2 dt\right) \tag{25}$$

where we used the moment generating function of Gaussian distribution and we denoted the variance of the noise $\xi_{m,i}^{\alpha}$ as $2\sigma_{\alpha}^2 \Delta t$. By putting them together, we have

$$P(\{x_{m,i}, a_{m,i}\}_{m,i}) \xrightarrow{\Delta t \to 0, M \to \infty} P(\{x_i(t), a_i(t)\}_i)$$

$$= \prod_{\alpha \in \{x,a\}} \prod_{i=1}^{N} \int \exp\left(\int k_i^{\alpha}(t)(\dot{\alpha}_i(t) - C_i^{\alpha}(t))dt + \int \left(k_i^{\alpha}(t)\right)^2 \sigma_{\alpha}^2 dt\right) \frac{dk_i^{\alpha}(t)}{2\pi i}. \tag{26}$$

Eliminating the formally introduced Gaussian noise by putting $\sigma_x = \sigma_a = 0$, we obtain

$$P(\{x_i(t), a_i(t)\}_i) = \int \mathcal{D}\tilde{\mathbf{x}}\mathcal{D}\tilde{\mathbf{a}} \exp\left(\tilde{\mathbf{x}}^T\left(\dot{\mathbf{x}} - (-\mathbf{x} + \mathbf{a} + \mathbf{J}\boldsymbol{\phi}(\mathbf{x}) + I)\right)\right.$$

$$\left. + \tilde{\mathbf{a}}^T\left(\dot{\mathbf{a}} - (-\boldsymbol{\Gamma}\mathbf{a} + \mathbf{B}\mathbf{x})\right)\right), \tag{27}$$

where, $\tilde{x}_i(t) = k_i^x(t)$, $\tilde{a}_i(t) = k_i^a(t)$, $\mathbf{J} = (J_{ij})$, $\prod_{m=1}^{M} \prod_{i=1}^{N} \int \frac{d\tilde{x}_{m,i}}{2\pi i} \to \int \mathcal{D}\tilde{\mathbf{x}}$, and $\prod_{m=1}^{M} \prod_{i=1}^{N} \int \frac{d\tilde{a}_{m,i}}{2\pi i} \to \int \mathcal{D}\tilde{\mathbf{a}}$. Here, $\boldsymbol{\Gamma}$ and $\mathbf{B}$ are the diagonal matrices whose $i$th diagonal component is given by $\gamma_i$ and $\beta_i$, respectively. Bold lowercase letters mean vectors whose inner product shall be simultaneously in the spatial and temporal directions, i.e., $\tilde{\mathbf{x}}^T\mathbf{x} = \sum_i \int \tilde{x}_i(t)x_i(t)dt$.

The moment generating functional of the stochastic process given by the probability distribution Eq (27) is defined as

$$Z[\mathbf{j}_x(t), \mathbf{j}_a(t), \tilde{\mathbf{j}}_x(t), \tilde{\mathbf{j}}_a(t)] := \int \mathcal{D}\mathbf{a}\mathcal{D}\mathbf{x} P(\{x_i(t), a_i(t)\}_i) \exp\left(\mathbf{j}_x^T\mathbf{x} + \mathbf{j}_a^T\mathbf{a}\right) \exp\left(\tilde{\mathbf{j}}_x^T\tilde{\mathbf{x}} + \tilde{\mathbf{j}}_a^T\tilde{\mathbf{a}}\right), \tag{28}$$

where $\mathbf{j}_x(t)$ and $\mathbf{j}_a(t)$ are the auxiliary variables corresponding to $x(t)$ and $a(t)$, respectively. Other auxiliary variables $\tilde{\mathbf{j}}_x(t)$ and $\tilde{\mathbf{j}}_a(t)$ are introduced to represent arbitrariness of their initial values [28]. We also introduced vector notations for integrals, $\int_{\infty}^{\infty} \prod_{m=1}^{M} \prod_{i=1}^{N} dx_{m,i} = \int \mathcal{D}\mathbf{x}$ and $\int_{-\infty}^{\infty} \prod_{m=1}^{M} \prod_{i=1}^{N} da_{m,i} = \int \mathcal{D}\mathbf{a}$.

Then, by putting Eq (27) to Eq (28), we have

$$Z[\mathbf{j}_x, \tilde{\mathbf{j}}_x, \mathbf{j}_a, \tilde{\mathbf{j}}_a](\mathbf{J}, \boldsymbol{\Gamma}) = \int \mathcal{D}\mathbf{a}\mathcal{D}\tilde{\mathbf{a}}\mathcal{D}\mathbf{x}\mathcal{D}\tilde{\mathbf{x}} \exp\left[S_{xa}[\mathbf{x}, \tilde{\mathbf{x}}, \mathbf{a}, \tilde{\mathbf{a}}](\boldsymbol{\Gamma}) - \tilde{\mathbf{x}}^T(\mathbf{J}\boldsymbol{\phi}(\mathbf{x}) + I)\right]$$

$$\times \exp\left[\mathbf{j}_x^T\mathbf{x} + \tilde{\mathbf{j}}_x^T\tilde{\mathbf{x}} + \mathbf{j}_a^T\mathbf{a} + \tilde{\mathbf{j}}_a^T\tilde{\mathbf{a}}\right] \tag{29}$$

$$S_{xa}[\mathbf{x}, \tilde{\mathbf{x}}, \mathbf{a}, \tilde{\mathbf{a}}](\mathbf{\Gamma}) = \tilde{\mathbf{x}}^T(\dot{\mathbf{x}} - (\mathbf{a} - \mathbf{x})) + \tilde{\mathbf{a}}^T(\dot{\mathbf{a}} - (\mathbf{Bx} - \mathbf{\Gamma a}))$$
$$= \sum_i \tilde{x}_i(\dot{x}_i - (a_i - x_i)) + \tilde{a}_i(\dot{a}_i - (\beta_i x_i - \gamma_i x_i)). \tag{30}$$

Since we want to know the network's behavior independent of each realization of the coupling matrix $\mathbf{J}$, let us average the moment generating functional over the distribution of the coupling matrix (quenched average). The coupling strengths independently follow the same normal distribution with mean 0 and variance $g^2/N$, and $\mathbf{J}$ appears in Eq (29) in the form of $\exp\left[\tilde{\mathbf{x}}^T\mathbf{J}\boldsymbol{\phi}(\mathbf{x})\right]$. So, we can average this as:

$$\prod_{i,j} \int dJ_{ij}\mathcal{N}(0, g^2/N; J_{ij}) \exp\left(\tilde{x}_i^T J_{ij}\phi(x_j)\right)$$

$$\propto \prod_{i,j} \exp\left(\frac{g^2}{2N}\left(\tilde{x}_i^T\phi(x_j)\right)^2\right) \tag{31}$$

$$= \exp\left(\sum_{i,j}\frac{g^2}{2N}\left(\int\int \tilde{x}_i(t)\tilde{x}_i(t')\phi(x_j(t))\phi(x_j(t'))dtdt'\right)\right)$$

$$= \exp\left(\frac{1}{2}\int\int\left(\sum_i \tilde{x}_i(t)\tilde{x}_i(t')\right)\left(\frac{g^2}{N}\sum_j \phi(x_j(t))\phi(x_j(t'))\right)dtdt'\right.$$

$$\left.-\sum_k \frac{g^2}{2N}\left(\int\int \tilde{x}_k(t)\tilde{x}_k(t')\phi(x_k(t))\phi(x_k(t'))dtdt'\right)\right)$$

$$\approx \exp\left(\frac{1}{2}\int\int\left(\sum_i \tilde{x}_i(t)\tilde{x}_i(t')\right)\left(\frac{g^2}{N}\sum_j \phi(x_j(t))\phi(x_j(t'))\right)dtdt'\right), \tag{32}$$

where we used the identity $\int dx\mathcal{N}(\mu, \sigma^2; x)\exp(ax) \propto \exp(\mu a + \frac{1}{2}a^2\sigma^2)$ in the second line. In the last line, we omitted the diagonal term, i.e., the second term of the exponential function, because this term scales in the order of $N$, which is sufficiently small compared to the off-diagonal first term, which scales with $N^2$ in the limit of $N \to \infty$. (Note that this difference in system size dependence will be important when discussing each neuron's intrinsic heterogeneity.) Using Eq (32), we obtain

$$\bar{Z}[\mathbf{j}_x, \tilde{\mathbf{j}}_x, \mathbf{j}_a, \tilde{\mathbf{j}}_a](\mathbf{\Gamma}) := \int\prod_{i,j}dJ_{ij}\mathcal{N}(0, g^2/N; J_{ij})Z[\mathbf{j}_x, \tilde{\mathbf{j}}_x, \mathbf{j}_a, \tilde{\mathbf{j}}_a](\mathbf{J}, \mathbf{\Gamma})$$

$$\approx \int\mathcal{D}\mathbf{a}\mathcal{D}\tilde{\mathbf{a}}\mathcal{D}\mathbf{x}\mathcal{D}\tilde{\mathbf{x}}\exp\left[S_{xa}[\mathbf{x}, \tilde{\mathbf{x}}, \mathbf{a}, \tilde{\mathbf{a}}](\mathbf{\Gamma}) + \mathbf{j}_x^T\mathbf{x} + \tilde{\mathbf{j}}_x^T\tilde{\mathbf{x}} + \mathbf{j}_a^T\mathbf{a} + \tilde{\mathbf{j}}_a^T\tilde{\mathbf{a}}\right]$$

$$\times\exp\left[\frac{1}{2}\int\int\left(\sum_i \tilde{x}_i(t)\tilde{x}_i(t')\right)\left(\frac{g^2}{N}\sum_j \phi(x_j(t))\phi(x_j(t'))\right)dtdt'\right], \tag{33}$$

where, for simplicity, we assumed that there is no external input, i.e., $I = 0$. Note that each second-order term of $\tilde{x}_i$ in the exponential of the above equation, i.e., the term of $\tilde{x}_i(t)\tilde{x}_i(t')$, shares the common factor $\frac{g^2}{N}\sum_j \phi(x_j(t))\phi(x_j(t'))$, which is independent of the subscript $i$. As we will see below, this independence from the neuron's index is key to deriving a single effective equation.

To proceed, let us define the auxiliary field $Q_1$:

$$Q_1(t,t') := \frac{g^2}{N} \sum_j \phi(x_j(t)) \phi(x_j(t')) \tag{34}$$

and represent this relationship by using the delta functional with an additional auxiliary field $Q_2$:

$$\delta\left[ -\frac{N}{g^2} Q_1(t,t') + \sum_j \phi(x_j(t)) \phi(x_j(t')) \right]$$

$$= \int \mathcal{D}Q_2 \exp\left[ \int \int \left( -\frac{N}{g^2} Q_1(t,t') Q_2(t,t') + \sum_j \phi(x_j(t)) Q_2(t,t') \phi(x_j(t')) \right) dt dt' \right]. \tag{35}$$

Putting this delta functional into Eq (33) allows us to treat $Q_1$ as an independent variable in the integral. Then, by using the Fourier representation of the delta functional, we have

$$\int \mathcal{D}Q_1 \delta\left[ -\frac{N}{g^2} Q_1(t,t') + \sum_j \phi(x_j(t)) \phi(x_j(t')) \right] \bar{Z}[\mathbf{j}_x, \tilde{\mathbf{j}}_x, \mathbf{j}_a, \tilde{\mathbf{j}}_a](\mathbf{\Gamma})$$

$$= \int \mathcal{D}Q_1 \mathcal{D}Q_2 \mathcal{D}\mathbf{a} \mathcal{D}\tilde{\mathbf{a}} \mathcal{D}\mathbf{x} \mathcal{D}\tilde{\mathbf{x}} \exp\left[ S_{xa}[\mathbf{x}, \tilde{\mathbf{x}}, \mathbf{a}, \tilde{\mathbf{a}}](\mathbf{\Gamma}) + \mathbf{j}_x^T \mathbf{x} + \tilde{\mathbf{j}}_x^T \tilde{\mathbf{x}} + \mathbf{j}_a^T \mathbf{a} + \tilde{\mathbf{j}}_a^T \tilde{\mathbf{a}} \right]$$

$$\times \exp\left[ \int \int \left( \sum_i \frac{1}{2} \tilde{x}_i(t) Q_1(t,t') \tilde{x}_i(t') + \sum_j \phi(x_j(t)) Q_2(t,t') \phi(x_j(t')) \right. \right.$$

$$\left. \left. -\frac{N}{g^2} Q_1(t,t') Q_2(t,t') \right) dt dt' \right]. \tag{36}$$

We can see that interactions among neurons are replaced by the effective interaction between each neuron and the auxiliary field. This result allows us to decouple neural dynamics, except for the effective interaction via the auxiliary field.

To have the explicit expression of the decoupled neural dynamics, formally rewrite the above expression by removing vector response terms $\mathbf{j}_x$, $\tilde{\mathbf{j}}_x$, $\mathbf{j}_a$, and $\tilde{\mathbf{j}}_a$ and introducing two scalar variables $j_{Q_1}$ and $\tilde{j}_{Q_2}$ to incorporate terms for the auxiliary field $j_{Q_1}^T Q_1 + \tilde{j}_{Q_2}^T Q_2$ into the generating functional [28]:

$$\bar{Z}[j_{Q_1}, \tilde{j}_{Q_2}] = \int \mathcal{D}Q_1 \mathcal{D}Q_2 \exp\left( S[Q_1, Q_2] + j_{Q_1}^T Q_1 + \tilde{j}_{Q_2}^T Q_2 \right), \tag{37}$$

where

$$S[Q_1, Q_2] = -\frac{N}{g^2} Q_1^T Q_2 + \sum_i \ln Z_i[Q_1, Q_2]$$

$$Z_i[Q_1, Q_2] = \int \mathcal{D}a_i \mathcal{D}\tilde{a}_i \mathcal{D}x_i \mathcal{D}\tilde{x}_i \exp\left( \tilde{x}_i \left( \dot{x}_i - (a_i - x_i) \right) + \tilde{a}_i \left( \dot{a}_i - (\beta x_i - \gamma_i a_i) \right) \right. \tag{38}$$

$$\left. + \frac{1}{2} \tilde{x}_i^T Q_1 \tilde{x}_i + \phi(x_i)^T Q_2 \phi(x_i) \right)$$

Here, we introduced notations $Q_1^T Q_2 = \int \int Q_1(t,t') Q_2(t,t') dt dt'$ and $\tilde{x}_i^T Q_1 \tilde{x}_i = \int \int \tilde{x}_i(t)^T Q_1(t,t') \tilde{x}_i(t') dt dt'$.

The exponent of the exponential function of Eq (37) increases with the order of $N$ as we increase the number of neurons in the network. Using this property, one can evaluate the above integral by using the saddle point approximation that allows us to replace the integral of the exponential function with the exponential function itself of the maximum values for its variables $Q_1^*$ and $Q_2^*$. The maximum condition, i.e., the saddle-point equation, is given by

$$\frac{\delta S[Q_1, Q_2]}{\delta Q_{\{1,2\}}} = 0 \tag{39}$$

where $\frac{\delta}{\delta Q_{\{1,2\}}}$ denotes the derivative of a functional in $Q_1$ and $Q_2$. Because these saddle-point equations are explicitly written as

$$\frac{\delta S[Q_1, Q_2]}{\delta Q_1(t, t')} = -\frac{N}{g^2} Q_2(t, t') + \sum_i \frac{1}{Z_i[Q_1, Q_2]} \int \mathcal{D}\tilde{x}_i \frac{1}{2} \tilde{x}_i(t) \tilde{x}_i(t') = 0$$

$$\frac{\delta S[Q_1, Q_2]}{\delta Q_2(t, t')} = -\frac{N}{g^2} Q_1(t, t') + \sum_i \frac{1}{Z_i[Q_1, Q_2]} \int \mathcal{D}x_i \phi(x_i(t)) \phi(x_i(t')) = 0,$$

we can solve them as

$$Q_1^*(t, t') = g^2 C_\phi$$
$$Q_2^*(t, t') = \frac{1}{2} \langle \tilde{x}_i(t) \tilde{x}_i(t') \rangle_{Q^*} = 0 \tag{40}$$
$$C_\phi(t, t') := \frac{1}{N} \sum_i \langle \phi(x_i(t)) \phi(x_i(t')) \rangle_{Q^*},$$

where $\langle \cdot \rangle_{Q^*}$ means the average with respect to the probability distribution given by the solution of the saddle-point equation. Then, using the solution, we have the final expression of the averaged moment generating functional:

$$\bar{Z}^* \propto \int \mathcal{D}\mathbf{a} \mathcal{D}\tilde{\mathbf{a}} \mathcal{D}\mathbf{x} \mathcal{D}\tilde{\mathbf{x}} \exp \left( S_{xa}[\mathbf{x}, \tilde{\mathbf{x}}, \mathbf{a}, \tilde{\mathbf{a}}](\mathbf{\Gamma}) + \sum_i \frac{g^2}{2} \tilde{x}_i^T C_\phi \tilde{x}_i \right). \tag{41}$$

The final expression is same to the moment generating functional with noise $\sigma_x = g^2 C_\phi$ and no coupling $\mathbf{J} = 0$ in Eq (26). Therefore it represents that each variable $x_i$ is commonly driven by the Gaussian process with mean 0 and the autocorrelation $g^2 C_\phi$. Thus, pulling back the expression to differential equations, we have the effective dynamics of the population of neurons driven by the Gaussian mean field:

$$\dot{x}_i(t) = -x_i(t) + a_i(t) + \eta_\phi(t) + I_i(t)$$
$$\dot{a}_i(t) = -\gamma_i a_i(t) + \beta_i x_i(t), \tag{42}$$

which is equal to Eq (4) of the main text.

## What happens if we naively average out intrinsic heterogeneity of neurons

In the previous section, we derived Eq (33) by averaging the heterogeneity in the coupling strengths $\mathbf{J} = (J_{ij})$. However, we left the intrinsic heterogeneity, namely the variability in each neuron's parameter $\gamma_i$, untouched. In this subsection, we will illustrate how the analysis

becomes problematic if we average the intrinsic heterogeneity in the same way as the coupling strengths.

Let us revisit Eqs (29) and (30), which provide the moment generating functional of the stochastic process before the averaging. Assuming that each neuron's parameter $\gamma_i$ independently follows a Gaussian distribution $\mathcal{N}(\mu_\gamma, \sigma_\gamma^2)$, we have the functional averaged over the intrinsic parameter as:

$$\bar{Z} = \int \prod_i d\gamma_i \mathcal{N}(\mu_\gamma, \sigma_\gamma^2; \gamma_i) Z[\mathbf{j}_x, \tilde{\mathbf{j}}_x, \mathbf{j}_a, \tilde{\mathbf{j}}_a] \tag{43}$$

where $Z[\mathbf{j}_x, \tilde{\mathbf{j}}_x, \mathbf{j}_a, \tilde{\mathbf{j}}_a]$ is the functional given by Eq (28). Since $\gamma_i$ is in only $S_{xa}$ of the functional, as described in Eq (30), performing the integral requires evaluation of

$$\bar{Z} = Z'[\mathbf{j}_x, \tilde{\mathbf{j}}_x, \mathbf{j}_a, \tilde{\mathbf{j}}_a] \int \prod_i d\gamma_i \mathcal{N}(\mu_\gamma, \sigma_\gamma^2; \gamma_i) \exp\left(-\gamma_i \tilde{a}_i^T a_i + \mu_\gamma \tilde{a}_i^T a_i\right), \tag{44}$$

where $Z'$ is equivalent to $Z$ except $S_{xa}$ in it is replaced by $\bar{S}_{xa} := \sum_i \tilde{x}_i^T (\dot{x}_i - (a_i - x_i)) + \tilde{a}_i^T (\dot{a}_i - (\beta x_i - \mu_\gamma a_i))$. We can perform the integral and obtain

$$\int \prod_i d\gamma_i \mathcal{N}(\mu_\gamma, \sigma_\gamma^2; \gamma_i) \exp\left(-\gamma_i \tilde{a}_i^T a_i + \mu_\gamma \tilde{a}_i^T a_i\right)$$
$$\propto \exp\left(\frac{\sigma_\gamma^2}{2}\left(\sum_i \int \int \tilde{a}_i(t) a_i(t) \tilde{a}_i(t') a_i(t') dt dt'\right)\right), \tag{45}$$

where we used the identity $\int dx \mathcal{N}(\mu, \sigma^2; x) \exp(ax) \propto \exp(\mu a + \frac{1}{2} a^2 \sigma^2)$ in the second line and the notation $a^T b = \int a(t) b(t) dt$.

The equation above corresponds to Eq (31) in the previous subsection. In that subsection, one could omit the order $N$ term because the leading order of the exponent in the exponential function was $N^2$. However, the order $N$ term cannot be neglected here because no higher-order terms are present. Consequently, instead of $\sum_j g^2 \phi(x_j(t)) \phi(x_j(t'))/N$, the factor $\sigma_\gamma^2 a_i(t) a_i(t')/2$ determines the variance of the effective Gaussian noise in the decoupled equation of neural dynamics. Unlike $\sum_j \phi(x_j(t)) \phi(x_j(t'))/N$, however, this factor depends on the neuron index $i$, making the resulting one-body equation heterogeneous. As a result, a single effective equation cannot be derived, and the effective equation must be given as a set of equations, still with the heterogeneity of each neuron. (Note that a single equation could be obtained if we were to forcibly replace $a_i(t) a_i(t')$ with $a(t) a(t')$, ignoring the $i$-dependence. However, as shown in Sect 3, this approach fails to produce accurate predictions.)

## Derivation of the relationship between power spectrums

In this subsection, we explain details of the derivation of the Eq (11) of the main text.

Let us define the Fourier transform $\mathcal{F}[\cdot]$ and the normalized Fourier transform $\mathcal{F}^L[\cdot]$ by

$$Y(\omega) = \mathcal{F}[y(t)](\omega) := \int_{-\infty}^{\infty} y(t) \exp(-i\omega t) dt \tag{46}$$

$$Y^L(\omega) = \mathcal{F}^L[y(t)](\omega) := \frac{1}{\sqrt{L}} \int_{-L/2}^{L/2} y(t) \exp(-i\omega t) dt. \tag{47}$$

Then, the effective network dynamics, Eq (4) of the main text, without external input

$$
\begin{aligned}
\dot{x}_i(t) &= -x_i(t) + a_i(t) + \eta_\phi(t) \\
\dot{a}_i(t) &= -\gamma_i a_i(t) + \beta_i x_i(t)
\end{aligned}
\tag{48}
$$

are transformed to

$$
\begin{aligned}
i\omega X_i^L(\omega) &= -X_i^L(\omega) + A_i^L(\omega) + H_\phi^L(\omega) \\
i\omega A_i^L(\omega) &= -\gamma_i A_i^L(\omega) + \beta_i X_i^L(\omega).
\end{aligned}
\tag{49}
$$

From the above expression, we have

$$
A_i^L = \frac{\beta_i}{i\omega + \gamma_i} X_i^L = \frac{(-i\omega + \gamma_i)\beta_i}{\omega^2 + \gamma_i^2} X_i^L,
\tag{50}
$$

where we have omitted $\omega$ from the expression for simplicity. Substituting this to the first line of Eq (49) gives

$$
\left( i\omega \left( 1 + \frac{\beta_i}{\omega^2 + \gamma_i^2} \right) + 1 - \frac{\gamma_i \beta_i}{\omega^2 + \gamma_i^2} \right) X_i^L = H_\phi^L.
\tag{51}
$$

Then, by multiplying complex conjugates of themselves to both sides of this expression, we have

$$
\begin{aligned}
\left\{ \omega^2 \left( 1 + \frac{\beta_i}{\omega^2 + \gamma_i^2} \right)^2 + \left( 1 - \frac{\gamma_i \beta_i}{\omega^2 + \gamma_i^2} \right)^2 \right\} X_i^L \bar{X}_i^L &= H_\phi^L \bar{H}_\phi^L \\
\frac{\omega^4 + (\gamma_i^2 + 1)\omega^2 + \gamma_i^2 + \beta_i^2 + \beta_i \omega^2 - 2\beta_i \gamma_i}{\omega^2 + \gamma_i^2} X_i^L \bar{X}_i^L &= H_\phi^L \bar{H}_\phi^L \\
\frac{1}{G(\omega; \gamma_i, \beta_i)} X_i^L \bar{X}_i^L &= H_\phi^L \bar{H}_\phi^L \\
S_{x_i}(\omega) &= G(\omega; \gamma_i, \beta_i) S_H(\omega),
\end{aligned}
\tag{52}
$$

where, in the last line, we used $S_{x_i}(\omega) = \lim_{L\to\infty} \left\langle X_i^L \bar{X}_i^L \right\rangle$ and $S_H(\omega) = \lim_{L\to\infty} \left\langle H_\phi^L \bar{H}_\phi^L \right\rangle$ that directly followed from the definition of the power-spectrum density.

For the right-hand side of the above equation, we have $S_H(\omega) = \mathcal{F}[\langle \eta(t)\eta(t-\tau)\rangle]$ from the Wiener-Khinchin theorem, and, using them, we can show

$$
\begin{aligned}
S_H(\omega) &= \mathcal{F}[\langle \eta(t)\eta(t-\tau)\rangle] \\
&= \mathcal{F}[\frac{g^2}{N} \sum_{i=1}^{N} \langle \phi(x_i(t))\phi(x_i(t-\tau))\rangle] \\
&= \frac{g^2}{N} \sum_{i=1}^{N} \mathcal{F}[\langle \phi(x_i(t))\phi(x_i(t-\tau))\rangle] \\
&= \frac{g^2}{N} \sum_{i=1}^{N} S_{\phi_i}(\omega) \\
&= g^2 \bar{S}_\phi(\omega),
\end{aligned}
\tag{53}
$$

with defining $S_{\phi_i}(\omega) := \mathcal{F}[\langle \phi(x_i(t))\phi(x_i(t-\tau))\rangle]$ and $\bar{S}_\phi(\omega) := \frac{1}{N}\sum_{i=1}^{N} S_{\phi_i}(\omega)$.

Substituting Eq (53) to Eq (52) and averaging both hand sides of the equation over the index $i$ gives the desired relationship between power-spectrums:

$$\bar{S}_x(\omega) := \frac{1}{N}\sum_{i=1}^{N} S_{x_i}(\omega)$$

$$= g^2\bar{S}_\phi(\omega)\left(\frac{1}{N}\sum_{i=1}^{N} G(\omega;\gamma_i,\beta_i)\right) \tag{54}$$

$$\rightarrow g^2\bar{S}_\phi(\omega)\bar{G}(\omega). \tag{55}$$

In the last line, we replace the arithmetic average of $G$ with its mean, that is, $\frac{1}{N}\sum_{i=1}^{N}$ $G(\omega;\gamma_i,\beta_i) \rightarrow \langle G(\omega;\gamma,\beta)\rangle_{\gamma,\beta} =: \bar{G}(\omega)$.

## Derivation of the equation for the transition point of the heterogeneous network

In this subsection, we will derive the equation, Eq (14), that determines the transition point $g_c$ of the network from the power-spectrum equation derived in the above subsection, Eq (55), or Eq (11) of the main text.

Assume that the activation function $\phi(x)$ satisfies the condition $|\phi(x)| \leq |x|$, which is the condition that is satisfied by most of the standard activation functions, including tanh and *ReLU*. Then, it directly follows that

$$\int |\phi(x_i(t))|^2 dt \leq \int |x_i(t)|^2 dt \tag{56}$$

$$\frac{1}{N}\sum_{i=1}^{N}\int |\phi(x_i(t))|^2 dt \leq \frac{1}{N}\sum_{i=1}^{N}\int |x_i(t)|^2 dt. \tag{57}$$

Because Parseval's theorem of the power spectrum gives

$$\int |x_i(t)|^2 dt = \frac{1}{2\pi}\int_{-\infty}^{\infty} |X_i(\omega)|^2 d\omega$$
$$\int |\phi(x_i(t))|^2 dt = \frac{1}{2\pi}\int_{-\infty}^{\infty} |\Phi_i(\omega)|^2 d\omega \tag{58}$$

Substituting them into Eq (57) and taking the mean for both hand sides of the equation gives

$$\frac{1}{N}\sum_{i=1}^{N}\frac{1}{2\pi}\int_{-\infty}^{\infty}\langle|\Phi_i(\omega)|^2\rangle d\omega \leq \frac{1}{N}\sum_{i=1}^{N}\frac{1}{2\pi}\int_{-\infty}^{\infty}\langle|X_i(\omega)|^2\rangle d\omega \tag{59}$$

$$\frac{1}{N}\sum_{i=1}^{N}\int_{-\infty}^{\infty} S_{\phi_i}(\omega)d\omega \leq \frac{1}{N}\sum_{i=1}^{N}\int_{-\infty}^{\infty} S_{x_i}(\omega)d\omega \tag{60}$$

$$\int_{-\infty}^{\infty}\bar{S}_\phi(\omega)d\omega \leq \int_{-\infty}^{\infty}\bar{S}_x(\omega)d\omega \tag{61}$$

Then, by substituting Eq (61) to Eq (55), we arrive

$$\int_{-\infty}^{\infty}\bar{S}_\phi(\omega)d\omega \leq g^2\int_{-\infty}^{\infty}\bar{S}_\phi(\omega)\bar{G}(\omega)d\omega. \tag{62}$$

Note that Eq (62) is an absolute inequality that must be satisfied by any values of $\bar{G}$, $g$, and $x$, as far as the activation function satisfies the above-introduced condition $|\phi(x)| \leq |x|$. Now, let assume that the coupling strength $g$ satisfies an inequality $\max_\omega g^2 \bar{G}(\omega) < 1$. Then, one can immediately see that $\bar{S}_\phi(\omega)$ must satisfies $\forall \omega, \bar{S}_\phi(\omega) = 0$ to fulfill the absolute inequality Eq (62). Because the power spectrum $\bar{S}_\phi(\omega) = 0$ for all $\omega$ means the neural activity is $\phi(x(t)) = 0$, we can conclude the network must be in the silent state when $\max_\omega g^2 \bar{G}(\omega) < 1$. On the other hand, if the coupling strength satisfies $\max_\omega g^2 \bar{G}(\omega) > 1$, we can say that the network is allowed to be in an active state, while we cannot say that the network must be in an active state. Thus, we can conclude that the condition

$$\max_\omega g_c^2 \bar{G}(\omega) = 1 \tag{63}$$

gives the critical coupling strength. Note that we can rewrite this condition as $\max_\omega g_c^2 \bar{G}(\omega = 0) = 1$ because $G(\omega; \gamma, \beta)$ takes its maximum value at $\omega = 0$ under the usual condition of $\gamma > \beta$ where the activity of the GPA neurons does not diverse.

## Analytical expression of the critical coupling strength for the two-point distribution of $\gamma_i$

In this subsection, we will give the analytical expression of the critical coupling strength $g_c$ when the decay rate $\gamma_i$ follows the two-point distribution:

$$\gamma_i = \begin{cases} \gamma_{low} & \text{probablity } p \\ \gamma_{high} & \text{probablity } 1 - p \end{cases}. \tag{64}$$

Let us assume that the model parameters fulfill the condition $\gamma_{high} > \gamma_{low} > \beta > 0$ to each single neural activity does not diverge. by differentiating Eq (8) of the main text, we have

$$G'(\omega) = \frac{-2\omega^5 - 4\gamma^2\omega^2 - 2\omega\left(\gamma^4 + 2\gamma^2\beta + \beta(2\gamma - \beta)\right)}{\left(\omega^4 + (\gamma^2 + 2\beta + 1)\omega^2 + (\gamma - \beta)^2\right)^2}. \tag{65}$$

Under the condition of $\gamma_{high} > \gamma_{low} > \beta > 0$, $G'(\omega) = 0$ has a unique real solution at $\omega = 0$. Therefore, since $G'(\omega = 0) = 0$ and $G'(\omega = +0) \leq 0$, it follows that $\omega = 0$ provides the maximum value of $G$ given by $G(0) = \frac{\gamma^2}{(\gamma - \beta)^2}$.

Then, as $\bar{G}(\omega)$ is just an average of $G(\omega)$ over the given distribution of $\gamma$ and $\omega = 0$ gives the maximum of $G$ regardless of values of $\gamma$, we can conclude that $\bar{G}(\omega)$ also takes its maximum value at $\omega = 0$. Therefore, combining this result with Eq (14) of the main text, we have the explicit expression of the transition point $g_c$ as follows:

$$\max_\omega g_c^2 \bar{G}(\omega) = g_c^2 \left\langle \frac{\gamma^2}{(\gamma - \beta)^2} \right\rangle_\gamma = 1 \tag{66}$$

$$g_c = \left( p \frac{\gamma_{low}^2}{(\gamma_{low} - \beta)^2} + (1 - p) \frac{\gamma_{high}^2}{(\gamma_{high} - \beta)^2} \right)^{-\frac{1}{2}} \tag{67}$$

## Supporting information

**S1 Appendix. Impact of continuous heterogeneity in intrinsic parameters on network dynamics and computational capacity.**
(PDF)

**S1 Fig. Network dynamics and temporal properties of neuron subgroups in a network with different decay rates.** (a) Temporal profiles of the activity of neurons with small (left) and large (right) decay rate parameters, corresponding to slower and faster time scales, respectively. (b) Autocorrelation functions of the same neuron groups are shown in (a). (c) Average autocorrelation functions within subgroups of neurons with ranges of decay rate parameters. (d) Distributions of relaxation times computed from the autocorrelation functions of individual neurons prior to averaging. Decay rates $\gamma_i$ follow a uniform distribution $U[1,10]$ across the network. Parameters $N = 5000$ and $g = 2.0$ are used.
(TIF)

**S2 Fig. Maximum power spectrum of network dynamics with continuous heterogeneity.** The left panel shows the result for a network in which the decay rates $\gamma_i$ follow a uniform distribution which center is 5. The right panel shows the result for a network in which the feedback strengths $\beta_i$ follow a truncated normal distribution with mean 0 and range $[-2,2]$. Red lines indicate the theoretical predictions.
(TIF)

**S3 Fig. Amplitudes of neuronal responses to a transient impulse input in a heterogeneous network.** The upper heatmap shows the response amplitudes of individual neurons to the same pulse input applied to the network. Neurons are sorted from top to bottom in ascending order of their feedback strength $\beta_i$. The lower panel shows the external input, which is shared by all neurons. The network operates in a stable regime where $g = 0.15 < g_c = 0.183$, and the feedback strengths $\beta_i$ follow a uniform distribution $U[0,2.9]$. Other parameters are $N = 3000$ and $\gamma = 3.0$.
(TIF)

**S4 Fig. Memory capacity (MC) of heterogeneous networks for various levels of heterogeneity.** Vertical dashed lines indicate the critical coupling strengths for each network. The input signal $u(t)$ is modeled as Gaussian white noise, and the time evolution of the network is computed using the Euler–Maruyama method for stochastic differential equations. The other parameters are $N = 3000$, $\gamma = 3$, $u(t) \sim \mathcal{N}(0, \sigma_{in}\sqrt{2dt})$, $\sigma_{in} = 0.1$, and $dt = 0.02$.
(TIF)

## Author contributions

**Conceptualization:** Futa Tomita, Jun-nosuke Teramae.

**Data curation:** Futa Tomita.

**Formal analysis:** Futa Tomita, Jun-nosuke Teramae.

**Funding acquisition:** Futa Tomita, Jun-nosuke Teramae.

**Investigation:** Futa Tomita, Jun-nosuke Teramae.

**Methodology:** Futa Tomita, Jun-nosuke Teramae.

**Project administration:** Jun-nosuke Teramae.

**Resources:** Jun-nosuke Teramae.

**Software:** Futa Tomita.

**Supervision:** Jun-nosuke Teramae.

**Validation:** Futa Tomita, Jun-nosuke Teramae.

**Visualization:** Futa Tomita.

**Writing – original draft:** Futa Tomita.

**Writing – review & editing:** Jun-nosuke Teramae.

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
