## [Decision Letter · Decision Letter 0]

21 May 2025

PCOMPBIOL-D-25-00641

Dynamical mean-field theory for a highly heterogeneous neural population with graded persistent activity of the entorhinal cortex

PLOS Computational Biology

Dear Dr. Teramae,

Thank you for submitting your manuscript to PLOS Computational Biology. After careful consideration, we feel that it has merit but does not fully meet PLOS Computational Biology's publication criteria as it currently stands. Therefore, we invite you to submit a revised version of the manuscript that addresses the points raised during the review process.

Please submit your revised manuscript within 60 days Jul 21 2025 11:59PM. If you will need more time than this to complete your revisions, please reply to this message or contact the journal office at ploscompbiol@plos.org. Please include the following items when submitting your revised manuscript:

We look forward to receiving your revised manuscript.

Kind regards,

Daniel Bush

Academic Editor

PLOS Computational Biology

Hugues Berry

Section Editor

PLOS Computational Biology

**Additional Editor Comments :**

The authors should consider simulating a wider range of time scales, clarifying the mean field equations, and exploring the use of chaotic dynamics to encode temporal information.

**Journal Requirements:**

At this stage, the following Authors/Authors require contributions: Futa Tomita. Please ensure that the full contributions of each author are acknowledged in the "Add/Edit/Remove Authors" section of our submission form.

4) Thank you for stating "The code used to run microscopic simulation and analysis is available at https://github.com/fuuta/GPADMFT_py." This link reaches a 404 error page. Please amend this to a new link or provide further details to locate the data.

5) Please ensure that the funders and grant numbers match between the Financial Disclosure field and the Funding Information tab in your submission form. Note that the funders must be provided in the same order in both places as well. Currently, the order of the grants is different in both places.

**Reviewers' comments:**

Reviewer's Responses to Questions

Reviewer #1: This paper presents an analytical method based on dynamical mean field theory and Fourier transform, and applies it to study the transition to chaos for randomly-connected neural networks where neurons have prolonged firing due to self-feedback mechanism. The authors show that when the network consists of two populations of neurons with long and short timescales for the self-feedback mechanism, the transition to chaos happens earlier (at lower coupling strength g). In addition, using the same method, the authors show that when the self-feedback timescales are sampled from a Gaussian distribution, the transition to chaos occurs later with higher values of sigma (the standard deviation of the Gaussian distribution), contrary to a previous result.

The technical aspect of the work is impressive, and the results seem important and relevant for researchers interested in how the type of slow neural activity observed in many parts of the brain could be beneficial for memory functions. On the other hand, I do think there are a few places where the paper could be made accessible for a wider audience:

1. The author mentioned briefly that chaotic activity (and therefore a larger fraction of slow neurons) could be useful in encoding temporal information. I think it would make the paper be of interest a wider audience if the authors could show this using an example. There have been proposals of how ramping cell and time cells can be used to encode temporal information (e.g. Shankar and Howard, Neural Computation 2002). There have been other proposals for network mechanisms that benefit temporal information encoding, for example using non-normal dynamics (e.g. Ganguli, Huh and Sompolinsky, PNAS 2008; Goldman, Neuron 2009). Both classes of models do not have chaotic activity. Rather they use the transient activity to encode temporal information before going back to the resting state. So I think it would be useful for readers to see how an alternative mechanism using chaos could also benefit temporal information processing.

2. I am not an expert of dynamic mean-field theory. While the authors did a good job stepping through the calculation, there are still a few places where I could not follow, and I believe it would benefit average audience if the authors could provide a bit more hand-holding in the calculation:

a. Equation 22: is P(\ksi^{\alpha}_{m, i}) the probability of x _{m, i} and a_{m, i}, given a particular realization of the noise \ksi? If so, I think it might be helpful to point this out explicitly or write it as P(x, a | \ksi)

b. Equation 22, second line: why is there an “i” in the “2*\pi*i” in the denominator? I didn’t remember the “i” from the Fourier expression of the delta function.

c. Equation 24: I didn’t get how the moment generating function Z can be transformed to the form k^2*\sigma*dt

d. Equation 35: why the signs in the exponential (+ \psi * Q2 * \psi – N/g^2 * Q1 * Q2) are reversed compared to equation 34?

e. Equation 39: I could not follow how are the extrema derived. It would be helpful to show the intermediate steps.

f. Text below Equation 40: why can we conclude from Equation 40 that each neuron is driven by a Gaussian process with mean 0 and autocorrelation C_{\psi}?

I want to emphasize that many of these questions may just be due to my lack of familiarity with DMFT, so the authors may choose to just respond here without changing the manuscript. But I believe this gives an idea of where an average reader might find difficult to follow. Based on this the authors may want to consider providing a bit more details and explanations for some parts of the mathematical derivation.

3. The distribution of the eigenvalues shown in Figure 5 is interesting. There seems to be a transition from 2 bulks to 3 bulks when p is increased. It would be interesting to study this a bit more. For example, do the eigenmodes in different bulks consist of neurons with different timescales? Is there a corresponding transition in the distribution of single-cell timescales (e.g. as calculated by the decay time constant of its autocorrelation) from a bi-modal distribution to a tri-modal one?

4. Line 351-352: the author mentioned that future work could look at the relationship between the dynamics of the network they studied, and the characteristic activity patterns observed in the entorhinal cortex. I think even for the current setup, there are something that could be examined. For example, are there single units in the network that show ramping activity or time-cell-like activity (localized temporal receptive field)? In addition, it is known that the time constants of time cells and ramping cells span a continuum. Do the timescales of single cell response in this network also form a continuum (this might relate to the eigenvalue distribution mentioned in point #3)?

Minor points

1. Fig 6: it would be more helpful for readers if legend was included in the figure.

2. Line 336-337: “external inputs do not decay” – I think the more accurate expression is “network activity triggered by external inputs do not decay”

3. Line 497: excepting � except

Reviewer #2: Present study proposes a computational method that allows one to apply dynamical mean-field theory to neural populations with different time scales. The problem is clearly articulated, and the proposed solution--averaging in Fourier space--is elegant. I find the analysis sound and emphasis on distinguishing intrinsic heterogeneity and connection heterogeneity bares important implications for future work.

My main concern is the simplification of the model to just two time-scale levels (high vs. low). Empirical recordings from the entorhinal cortext suggest a more continuous spectrum of time scales (e.g. Tsao et al, 2018 Nature; Bright et al., 2020, PNAS). It would be interesting to explore whether a transition point can be derived under such highly variable time scales. At minimum, a discussion of this point would be valuable. Ideally, a simulation could strengthen the papers appeal to a broader audience.

Minor comments:

1. The Github link does not work.

2. Several parameters are discussed throughout the paper but are not always referred consistently. It would make it easier to track parameters are referenced with both parameter name and its role in the function (e.g., coupling strength g) throughout, especially when a parameter is reintroduced several pages after its initial introduction.

**Have the authors made all data and (if applicable) computational code underlying the findings in their manuscript fully available?**

Reviewer #1: **No: **The authors have not provided the code for the simulation results in the paper, as well as the parameters for the networks analyzed.

Reviewer #2: Yes

PLOS authors have the option to publish the peer review history of their article (what does this mean?). If published, this will include your full peer review and any attached files.

Reviewer #1: No

Reviewer #2: No

**Figure resubmission:**
---

## [Decision Letter · Decision Letter 1]

3 Sep 2025

Dear Dr Teramae,

We are pleased to inform you that your manuscript 'Dynamical mean-field theory for a highly heterogeneous neural population with graded persistent activity of the entorhinal cortex' has been provisionally accepted for publication in PLOS Computational Biology.

Best regards,

Daniel Bush

Academic Editor

PLOS Computational Biology

Hugues Berry

Section Editor

PLOS Computational Biology

Reviewer #1:

Reviewer #2:

Reviewer's Responses to Questions

**Comments to the Authors:**

Reviewer #1: Thank you for the additional analyses and clarification. I do not have further concerns and recommend the manuscript for publication.

Reviewer #2: The authors have addressed my previous concern.

**Have the authors made all data and (if applicable) computational code underlying the findings in their manuscript fully available?**

Reviewer #1: Yes

Reviewer #2: Yes

PLOS authors have the option to publish the peer review history of their article (what does this mean?). If published, this will include your full peer review and any attached files.

Reviewer #1: No

Reviewer #2: **Yes: **Rui Cao

---

## [Editor Report · Acceptance letter]

PCOMPBIOL-D-25-00641R1

Dynamical mean-field theory for a highly heterogeneous neural population with graded persistent activity of the entorhinal cortex

Dear Dr Teramae,

I am pleased to inform you that your manuscript has been formally accepted for publication in PLOS Computational Biology. Your manuscript is now with our production department and you will be notified of the publication date in due course.

With kind regards,

Anita Estes
